# Estimating Long-Term Annual Energy Production from Shorter Time Series Data: Methods and Verification with a 10-Year Large-Eddy Simulation of a Large Offshore Wind Farm

Bernard F. A. Postema[1,2], Remco A. Verzijlbergh[1,3], Pim Van Dorp[1], Peter Baas[1], and Harm J. J. Jonker[1, 4]

[1]Whiffle BV, Molengraaffsingel 8, 2629 JD Delft, the Netherlands
[2]Meteorology and Air Quality Group, Wageningen University, Droevendaalsesteeg 3a 6708 PB, Wageningen, The Netherlands
[3]Department of Engineering Systems and Services, Delft University of Technology, Jaffalaan 5, 2628 BX Delft, The Netherlands
[4]Department of Geoscience and Remote Sensing, Delft University of Technology, Stevinweg 1, 2628 CN Delft, The Netherlands

**Correspondence:** Bernard F. A. Postema (bernard.postema@whiffle.nl)

**Abstract.**

Models used in wind resource assessment (WRA) range from engineering wake models and computational fluid dynamics models, to mesoscale weather models with wind farm parametrizations and, more recently, large-eddy simulation (LES). The latter two produce time series of wind farm power of a certain period. This simulation period is, in the case of LES, mostly

limited to $\leq 1$ year, due to the computational costs. However, estimates of long-term (O(10 y)) power production are of high value to many parties involved in WRA. To address the need to calculate long-term annual energy production from $\leq 1$ year model runs, therefore, this paper presents methods to estimate the long-term (O(10 y)) power production of a wind farm, using a $\leq 1$ year simulation. To validate the methods, 10 years of LES of a hypothetical large offshore wind farm are performed.

The methods work by estimating the conditional probability densities between wind farm power from the LES and wind

speed from reanalysis data (ERA5), from a short ($\leq 1$ year) LES run. The conditional probability densities are then integrated over 10 years of ERA5 wind speed, yielding an estimate of the long-term mean power production.

This 'long-term correction' method is validated on varying simulation periods, selected with four different day selection techniques. When applied to a simulation period of 365 consecutive days, the methods can estimate the 10 year mean power production with a mean absolute error of around 0.35 % of the long-term mean. When choosing the simulation period with

15 day-selection techniques that represent the long-term climate, only roughly 200 simulation days are needed to achieve the same accuracy.

Finally, a method to also include wind observations in the long-term correction is presented and tested. This requires an additional 'free stream' LES run without active turbines, and gives estimates of long-term power and wind that are corrected for a potential LES bias. Although validation of this final approach is difficult in the employed modeling strategy, it gives

valuable insights, and fits within the common WRA practice of combining models and observations.

The presented techniques are based on physical arguments, computationally cheap, and simple to implement. Furthermore, they are not limited to LES, but can be applied to other time series-based models. As such, they could be a useful extension to the diverse set of modeling, observational, and statistical techniques used in WRA.

## 1 Introduction

One of the main aims of wind resource assessment (WRA) is to model the power production of a wind farm given the broader environmental conditions. Such models vary greatly in their basic formulation, complexity, and computational costs; and as a consequence of those factors, they also vary in their application. Engineering wake models (Göçmen et al., 2016) or computational fluid dynamics models are typically used to produce power estimates given a certain 'flow case' (Locascio et al., 2022; Laan et al., 2022), after which the mean power is a weighted average of those flow cases. Contrastingly, there are models that produce that produce power production as time series, given the changing environmental conditions over some predefined period. Examples of these are weather models that include wind farm parametrizations, for example the one by Fitch et al. (2012) in the WRF mesoscale weather model. A more recent development is the use of atmospheric large-eddy simulation (LES) for these purposes. LES is a class of fine-scale computational fluid dynamics models, which resolution of about $10\,\mathrm{m}$ to $100\,\mathrm{m}$ allows explicit simulation of the most energetic part of the turbulence in the atmospheric boundary layer. Although its traditional application is to research fundamental meteorological processes (Stoll et al., 2020), LES can now be coupled to realistic weather data (Schalkwijk et al., 2015a; van Stratum et al., 2023), and can furthermore be used in WRA to understand and quantify the interactions between wind farms and the atmosphere in a 'real-weather' setting (Baas et al., 2023).

The current state of the art of using LES in WRA is to simulate one year, which comprises 365 consecutive days (e.g. the year of 2023), in order to provide an overlap with observation data, which are often only available during roughly one year. The simulation then includes the most important effects of weather conditions on wind park power production; on the turbulent, to the synoptic, to the seasonal timescale. This approach, however, means that any inter-annual variability is not taken into account. This is a major shortcoming, because the multi-year (O(10 y)) projected power production is of high interest to many parties involved in wind energy development, and interannual variability is significant (about 4 % of the long-term mean (Pryor et al., 2018)). Methods to accurately estimate long-term average power production without resorting to either a computationally cheaper model of lower physical fidelity, or a longer but too expensive LES run, would therefore be very useful in the WRA field.

To address this need, this work presents several methods to correct the power production and other (meteorological) variables as estimated by a short ($\leq 1$ y) LES run of a large offshore wind farm for interannual variability and/or bias with respect to observations, thus giving an estimate of the long-term (climatological) value of those variables. These hereby called 'long-term correction' estimate the long-term probability density of an LES variable, given *i)* the short-term joint probability density between the said LES variable and a reference variable in the reanalysis data, and *ii)* the long-term probability density of the reference variable in the reanalysis data. The motivation of using the presented methods is the prohibitive computational cost

of doing a O(10 y) LES run. Notwithstanding their suitability to LES in its current role in WRA, the methods can also be applied to any time series-based model, such as mesoscale models. In this study, LES results serve as an example application and validation of the presented long-term correction methods.

The presence or absence of observation data suggest the following three scenarios in which long-term correction can typically be applied:

1. LES wind farm power output of a predetermined set of 365 consecutive days is available, and needs to be corrected for interannual variability, but not for a bias with respect to observations.

2. The LES run period can be freely chosen, because there are no observations that need to be concurrent. The run period can therefore be a smartly chosen (representative) selection of days, which, after applying the long-term correction method, yields the best estimate of the long-term mean wind farm power.

3. LES wind farm power output of a predetermined set of 365 consecutive days is available. The power output needs to be long-term corrected for interannual variability and simultaneously be corrected for the bias with respect to observations.

Correcting or downscaling weather- or climate model data to better match observed reality is a widespread practice in environmental science. In this field, two main types can be distinguished (e.g.: Ekström et al., 2015; Holthuijzen et al., 2021): dynamical downscaling, in which a more accurate model is used to refine results from an often coarser model; and statistical downscaling, in which statistical relationships between the modeled and observed variable are used to correct the modeled variable. These methods generally produce corrected timeseries, but can include spatial dimensions as well (Holthuijzen et al., 2021). The current study uses aspects of both the dynamical and statistical type. Firstly, the data to be corrected itself come from an LES downscaling of ERA5. Secondly, and this is the core message of the study, the probability distributions are then corrected for the difference in weather conditions between the simulation period and the long-term climate. This is done by estimating the conditional probability density between the variable of interest and ERA5 wind from the short simulation, and then using this relationship to modify the probability density of the variable of interest to represent the long term. In this sense, the long-term correction method is most related to statistical downscaling, but instead of correcting for a model deficiency, the long-term correction method corrects for non-representative simulation periods.

In this study, 10 years of weather conditions and atmospheric flow through a large (960 MW) hypothetical wind farm on the North Sea are simulated with an LES, in order to validate the long-term correction methods in the three typical situations listed above. There is a long record of undisturbed observations in the location of the hypothetical wind farm, allowing validation of the LES, and testing the method of bias correction. The employed LES code is the Atmospheric Simulation Platform for Innovation, Research, and Education (ASPIRE), a modeling suite formerly often referred to with the name of its LES core: GRASP (GPU-Resident Atmospheric Simulation Platform). ASPIRE has its origins in the Dutch Atmospheric Large Eddy Simulation (DALES) model (Heus et al., 2010), but has since been ported to Graphics Processing Units (Schalkwijk et al.,

2012, 2015b), and is currently used for research as well as commercial purposes, mainly in the wind energy sector. Previous work with ASPIRE includes: Williams et al. (2024), Oldbaum (2019), Baas et al. (2023), Verzijlbergh (2021) (wind farm modeling); Schepers et al. (2021), Taschner et al. (2023) (turbine physics and loads); Gilbert et al. (2020), Alonzo et al. (2022)
(wind forecasting); Kantharaju et al. (2023), Storey and Rauffus (2024) (wind climate modeling); and Bieringer et al. (2021) (dispersion).

The present work focuses on the performance of the long-term correction methods given the employed model setup: validation of the wind farm modeling itself is not part of the scope. The paper is structured as follows: Sect. 2 gives the theory behind
the long-term correction methods, Sect. 3 describes the LES setup, methods of wind farm modeling, observation data, and day selection techniques. Then, results and discussion will be presented in Sect. 4 following the three scenario's above, and finally conclusions on the application of the long-term correction methods will be drawn in Sect. 5.

## 2  Theory of long-term correction of wind farm power production and wind

The goal of this work is to estimate long-term (O(10 y)) mean values of wind farm power production and wind from a much
shorter (O(1 y)) LES run. The mean values of that shorter run can deviate from their true long-term counterparts for two reasons:
*i)* the short LES run has different mean meteorological conditions than the long-term mean (this is normal climatological variability), and *ii)* the LES may display a consistent mean bias with respect to reality (this can be caused by a multitude of model flaws). The first can be corrected with purely atmospheric reanalysis data, such as the ERA5 reanalysis dataset, which is a global historical record of atmospheric conditions (Hersbach et al., 2020), and in this study also provides boundary
conditions for the LES. The second also needs on-site observations that are concurrent with the LES run, to quantify the bias. The following two sections explain the long-term correction methods designed to correct these deviations and to arrive at an estimate of the long-term mean power production and wind.

### 2.1  Long-term correction without observations

An LES run, which can be either a set of consecutive days or a smartly chosen sample (scenarios 1 and 2 in the introduction),
provides timeseries of power production of the entire wind farm ($P$), and, for instance, wind speed at a location in or close to the wind farm ($M$). These two variables are positive, random, and continuous, and have probability densities $f_L(P)$ and $g_L(M)$, where the subscript L refers to the LES.

Also, from an external atmospheric reanalysis dataset (in this case ERA5) the wind record during the simulation time can be taken, which has its own wind distribution $g_{ERA}(M)$. Furthermore, the *joint* probability densities between LES power and
ERA5 wind ($h_{L, ERA}(P, M)$) can be calculated. Total probability being unity requires that:

$$\int_0^\infty f_L(P)dP = \int_0^\infty g_{ERA}(M)dM = 1,$$ (1)

and (dropping the integration limits from now on):

$$\int\int h_{\text{L, ERA}}(P,M)dPdM = 1. \tag{2}$$

Also, integrating the joint probability density over one variable retrieves the marginal probability density of the other; for example:

$$\int h_{\text{L, ERA}}(P,M)dM = f_{\text{L}}(P). \tag{3}$$

Then, the *conditional* probability density is defined as:

$$h_{\text{L | ERA}}(P,M) = \frac{h_{\text{L, ERA}}(P,M)}{g_{\text{ERA}}(M)}, \tag{4}$$

which describes the distribution of the first variable between brackets, given a value of the second. From this definition, the classical result of Bayes' rule follows. Rewriting eqn. 4, and making use of the integration eqn. 3 gives an expression for $f_{\text{L}}(P)$:

$$\int h_{\text{L | ERA}}(P,M)g_{\text{ERA}}(M)dM = \int h_{\text{L, ERA}}(P,M)dM = f_{\text{L}}(P). \tag{5}$$

Denoting long-term, climatological, (O(10 y)) counterparts of the distributions with hats, eqn. 5 takes on the analogous form for the long-term:

$$\hat{f}_{\text{L}}(P) = \int \hat{h}_{\text{L, ERA}}(P,M)dM = \int \hat{h}_{\text{L | ERA}}(P,M)\hat{g}_{\text{ERA}}(M)dM. \tag{6}$$

If the LES run period is long enough and includes a sufficiently diverse range of weather conditions, it can be assumed that the conditional probability between power and wind approximates its long-term counterpart:

$$h_{\text{L | ERA}}(P,M) \approx \hat{h}_{\text{L | ERA}}(P,M). \tag{7}$$

Physically, this means assuming that the short simulation accurately captures the range of wind farm power production values that belong to a given wind speed. As will be shown in Sect. 4.1, this range can be significant.

Then, substituting this assumption in eqn. 6 gives an estimate of the long-term distribution of power production:

$$\hat{f}_{\text{L}}(P) \approx \int h_{\text{L | ERA}}(P,M)\hat{g}_{\text{ERA}}(M)dM. \tag{8}$$

So, eqn. 8 provides an estimate of the long-term probability distribution of LES power production (or wind, analogously) given *i)* the short-term conditional probability density of LES power (or wind) and reanalysis wind, and *ii)* the long-term probability distribution of the reanalysis wind. The long-term mean power production can then by calculated by taking the first moment of the distribution:

$$\overline{P} = \int f_{\mathrm{L}}(P)P dP, \tag{9}$$

and this mean power value (typically in MW) can be translated to the annual energy production (AEP, typically in GWh) by multiplying with one year.

## 2.2 Integrating observation data to correct for a model bias

In the practice of LES modeling, it is often found that the wind speed displays a mean bias of $O(0.1 \ \mathrm{m\,s^{-1}})$ with respect to observations (Kantharaju et al., 2023; Storey and Rauffus, 2024) (scenario 3). Because observations for WRA are usually done in an undisturbed environment (before construction of the wind farm), a second 'free stream' validation LES run is needed to determine this bias. Such a free stream run has no turbines included, or it has turbines that exert no force on the flow, and therefore leave it undisturbed. Apart from this, the two run setups are identical, allowing for a direct quantification of wake- and blockage losses.

The potential wind bias identified in a free stream run will also be similarly present in the wind farm power production with active turbines, and its long-term corrected counterpart, when applying the methodology of the previous section. To estimate long-term mean power production while also correcting for a bias, therefore, a different approach needs to be taken, which cannot rely on the absolute wind- or power values produced by the LES (because they are biased). Rather, this new approach needs to rely on the statistical relationship between power production of the LES run with active turbines, and the LES wind of a free stream run, i.e. $h_{\mathrm{L\,|\,FSL}}(P, M)$, where the subscript FSL denotes free stream LES, and the subscript L keeps referring to the run with active turbines. Despite its hypothetical nature (it cannot be measured in any way in reality), this conditional probability is assumed to be accurate, because of the explicit representation of many important fluid dynamical and meteorological processes that affect it; such as wakes, blockage, and stability effects (see e.g. Mehta et al., 2014; Breton et al., 2017).

An estimate of the long-term power distribution can then be obtained by integrating the product of $h_{\mathrm{L\,|\,FSL}}(P, M)$ and the probability distribution of unbiased long-term wind speed. The latter can be obtained from observations combined with reanalysis data. This is done in the form of a Measure-Correlate-Predict (MCP) procedure (a common technique used for WRA, see e.g. Carta et al. (2013)), which (per wind direction bin) fits the reanalysis wind to observation data, and then applies this fit to the long-term reanalysis record, thereby creating a semi-artificial long-term wind record, which has no bias with respect to the observation data from which it was constructed. This long-term MCP wind can therefore be seen as a bias-corrected version of the reanalysis wind. Using the long-term distribution of the MCP wind ($\hat{g}_{\mathrm{MCP}}(M)$) together with the conditional probability $h_{\mathrm{L\,|\,FSL}}(P, M)$ gives the estimate of the long-term power distribution the following form:

$\hat{f}(P) \approx \int h_{\mathrm{L}\,|\,\mathrm{FSL}}(P,M)\hat{g}_{\mathrm{MCP}}(M)dM.$            (10)

where the subscript L on the left hand side has been dropped, because the equation aims to estimate the real long-term power production, not its (possibly biased) long-term LES value. So, eqn. 10 (see the similarity to eqn. 8) provides a way to integrate observations into the long-term correction of power. An analogous form of eqn. 10 also works for non-free stream wind (wind disturbed by the wind farm, in the run with active turbines), but not for the free stream wind itself, because the integral then
reduces to $\hat{g}_{\mathrm{MCP}}(M)$.

The integral quantities presented until here are computed as discrete sums when applying long-term correction. This means the quantities of interest need to be binned in such a way that the assumption $h_{\mathrm{L}\,|\,\mathrm{ERA}}(P,M) \approx \hat{h}_{\mathrm{L}\,|\,\mathrm{ERA}}(P,M)$ holds as well as possible. The implied assumption is that all quantities that influence (either directly or indirectly) the power production
(stability, wind direction, air density) also obey this approximation. For wind, bins of $0.75 \mathrm{\ m\ s}^{-1}$ are chosen, and sensitivity to this value will be shown. If one is interested in the long-term mean of the variable that is being long-term corrected (in the previous examples power production), bins of that variable can be arbitrarily small. This is because in the final calculation of the long-term mean (for power: $\overline{P} = \int f_{\mathrm{L}}(P)P dP$), the bins are all aggregated. Furthermore, any point masses in the distribution (for wind farm power, zero and rated power are point masses) need to be at bin center values. This is to ensure that their (often
high) occurrence are assigned their correct point mass values.

## 3    Methods and data

### 3.1    LES setup

The employed LES code, named the Atmospheric Simulation Platform for Innovation, Research, and Education (ASPIRE) has gradually evolved from its root, the Dutch Atmospheric Large-Eddy Simulation (Heus et al., 2010). Key developments
were its porting to Graphics Processing Units (GPUs) (Schalkwijk et al., 2012, 2015b), coupling to reanalysis- or large-scale forecast weather data, and most recently, the transition from periodic boundary conditions to open boundary conditions. In the current setup, the core LES domain is nested in a coarser mesoscale-type simulation with a resolution of $1.5 \mathrm{\ km}$, which is directly coupled to ERA5 with open boundary conditions. This coarser simulation has the same model formulation as the LES, except for turbulence, which is completely parametrized. In this way, the model setup includes the basic elements of
meso-scale dynamics and gradients, such as fronts or land-sea transitions, which is problematic in a periodic simulation.
For this study, an LES domain (Fig. 1) of $30.72 \mathrm{\ km}$ by $30.72 \mathrm{\ km}$ and a horizontal resolution of $120 \mathrm{\ m}$ is chosen. The choice for this relatively coarse resolution (in the context of LES), is a pragmatic one. For the current purposes of presenting a 10-year simulation, it is considered to be a good trade-off between accuracy and computational cost. This was shown by Baas et al. (2023): refining their LES resolution to $60 \mathrm{\ m}$ has a relatively small effect on total aerodynamic losses of a $770 \mathrm{\ MW}$ wind farm.
Also in the present study, supplementary material is provided with a resolution study of the Horns Rev wind farm, with the

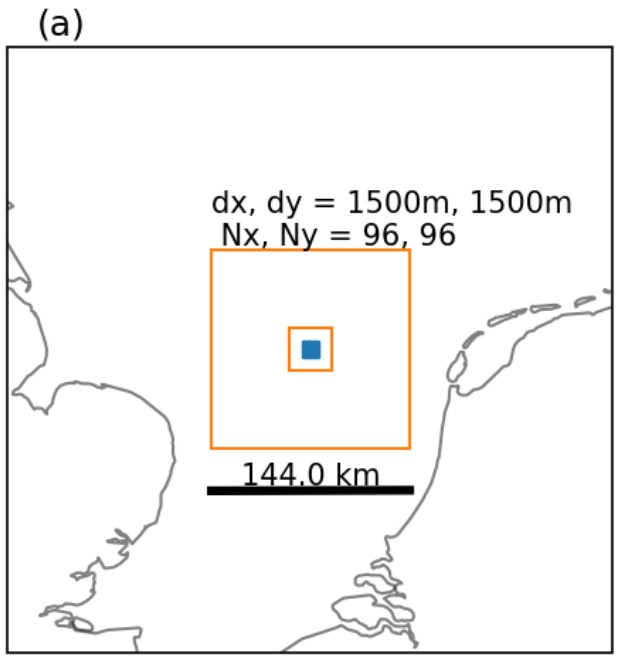
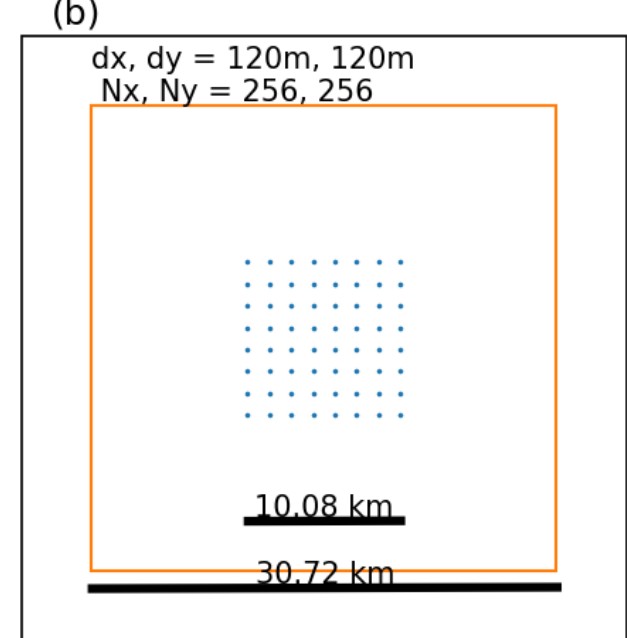

**Figure 1.** The simulation setup. a) the LES (small orange square) is nested in a coarser simulation (large orange square) on the North Sea. b) The LES domain with the wind farm layout (dots).

same modeling approach. Here too, it is shown that coarsening the resolution from 20 m to 120 m has a small effect on the essential physics of boundary layer meteorology and wind farm dynamics as represented in the LES.

Turbulence in the LES is generated by adding fluctuations to the inflow conditions from the mesoscale simulations. These fluctuations are themselves produced by a smaller periodic precursor LES that is driven by the mesocale field at the LES boundaries. Hence, these added fluctuations are consistent with atmospheric conditions (Storey and Rauffus, 2024). Sea surface roughness is parametrized according to Charnock (1955), and subgrid-scale turbulence is described by an anisotropic minimum dissipation turbulence parametrization (Verstappen, 2011; Rozema et al., 2015). In the current study, no microphysical parametrizations are used, and radiative tendencies are added from the ERA5. The vertical spacing of the 64 model levels starts at 30 m at the surface, and stretches exponentially to the domain top at 3 km. The coarser simulation around the LES has a domain of 144 km and a horizontal resolution of 1.5 km. Its 64 vertical levels start with a spacing of 40 m and stretch exponentially to the top at 8 km. The domains are on the southern North Sea, centered at 53.22°N, 3.22°E, and basic meteorological quantities are written at 10-minute intervals at that location. For the long-term correction, hourly averaged values will be used. The simulation period is 2010 - 2019, each day of which is simulated separately with a spin-up time of 2 hours.

## 3.2 Wind farm modeling

A hypothetical wind farm with a regularly spaced square layout of 8 by 8 turbines is included in the LES. The turbine type is the 15 MW offshore reference wind turbine of the International Energy Agency, described in Gaertner et al. (2020), and the turbines are spaced six times their rotor diameter ($6 \cdot 240$ m $= 1440$ m) (Fig. 1).

Turbines are implemented in the LES according an actuator disk model (Meyers and Meneveau, 2010; Calaf et al., 2010). In this approach, grid-specific power- and thrust coefficients of the turbine are first calculated offline from the manufacturer's information. During the simulation, the power production of each individual turbine as well as its force on the flow can be determined. In this way, turbines produce individual wakes and interact.

Two separate LES runs are performed for this study: one with active wind turbines, named the *realistic* run; and one where the wind turbines do not exert any force on the flow, named the *free stream* run. In the following, we will refer to wind and power production from those runs as realistic power or wind and free stream power or wind, respectively. The turbines in this free stream run have their thrust coefficient set to zero, meaning that they produce power, but no wakes; and therefore do not interact. A more elaborate description of the wind farm modeling can be found in Baas et al. (2023).

## 3.3 Observation data

10-minute wind anemometer data during 2010-2019 from the K13 offshore platform (53.22°N, 3.22°E, 75.3 m height) were obtained via the application programming interface of the Royal Netherlands Meterological Institue (KNMI) at https://dataplatform.knmi.nl/dataset/windgegevens-1-0 (last accessed 2 May 2024). The original data were unvalidated and contained periods when the wind speed erroneously decreases to zero. These were removed, together with all days that did not have complete data. These two cleaning steps removed 25 % of the data points.

## 3.4 Day selection techniques

A simple set of day selection techniques is applied to investigate the accuracy of the long-term correction methods as a function of LES run time (scenario 2). More involved techniques exist to find a sample of days that is representative of the wind climate (see e.g. Rife et al. (2013)), but their application is not the object of this study.

Day selection techniques may have an inherent source of randomness (such as simply selecting random days), and applying them many times gives an indication of the error statistics of the long-term correction method. Other methods are deterministic, and some form of spread or randomness needs to be introduced in order to gauge the error statistics of the long-term correction method. The following day selection techniques and their associated introduced source of spread are tested in this research:

- *consecutive*: a number of consecutive days are selected. Spread is introduced by starting on a random day, ensuring the sample falls within the time span of the LES.

- *random*: a number of random days are selected. Spread is introduced by using different random realizations.

- *ordered*: all days are ordered (sorted) based on their mean ERA5 100 m wind speed. Days are then equidistantly chosen from this series, in order to end up with the desired sample size. Spread among different samples is introduced by first excluding 365 random days from the 10 years, and then taking the equidistant sample.

- k-*means*: a standard *k*-means clustering method is applied on daily mean ERA5 values of zonal and meridional 100 m wind. Spread is introduced by first excluding 365 random days from the 10 years, and then applying the *k*-means algorithm.

## 4  Results and discussion

The following sections first describe the general performance of the realistic and free stream 10-year LES runs, in terms of wind statistics and comparison to observations. Then, as an illustration of the long-term correction method, Sect. 4.2 gives the illustration of using 2010 to estimate the mean power production of 2010-2019 as estimated by the LES. This leads the way to using many consecutive years to estimate the long-term mean LES power production (scenario 1), from which error metrics that describe the performance of the method can be calculated. Similar error metrics will be presented in Sect. 4.3, in which the long-term correction method will be applied to the different day selection techniques (scenario 2). Finally, Sect. 4.4 shows how observations can be integrated, which will give estimates of the (bias-free) real long-term mean power production (scenario 3).

### 4.1  Characterization and validation of the 10 y LES

Figure 2 shows basic validation statistics of the free stream LES run with observations and ERA5 wind. In general, the LES captures the wind conditions satisfactorily, despite its bias of -0.36 $\mathrm{m\,s^{-1}}$, which is typical in this application.

To illustrate the approach of the free stream and realistic run, Fig. 3 shows the joint (for the realistic run) and marginal probability density functions (PDFs) (for both runs) of power production of the wind farm and wind in the center of the domain at 75.3 m height. The joint PDF can be read as a probabilistic wind farm power curve, i.e.; it shows the mapping from wind conditions to power production. This mapping is not unique: for a given wind speed in the center of the park, there is a considerable range of power production values that are plausible. This reflects the combined effects of other factors than wind, such as stability and wind direction, that indirectly influence the power production through their effects on the turbine wakes. The relative difference in mean power production between the free stream and realistic run represents all internal aerodynamic losses and has a value of 17.5 %. This is mainly due to lower occurrence of rated power conditions in the realistic run. Wind conditions are similarly affected by including realistic turbines in the LES: the mean wind decreases by 19 %, and the wind PDF loses its characteristic Weibull shape.

### 4.2  Long-term correction of 365 consecutive days for climatological variability

In this section, the performance of the long-term correction method of 365 days to 10 years will be described. This entails evaluating the integrals in eqns. 8 and 9 and is an application of scenario 1 as described in the introduction. Before statistically

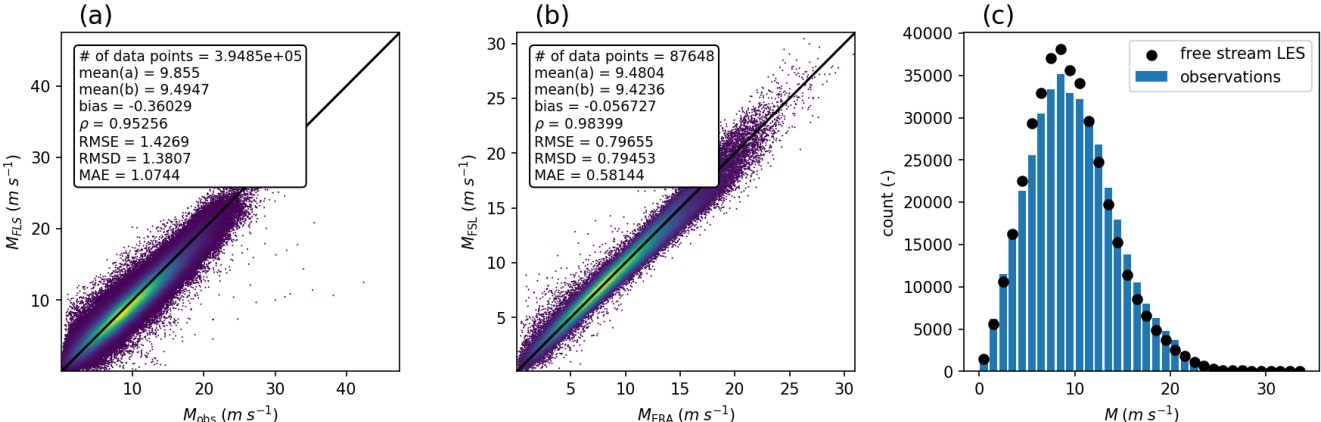

**Figure 2.** Basic validation of the free stream 10-year LES run. a) 10-minute free stream LES wind speed against observed wind speed at 75.3 m and standard error metrics, b) 1-h free stream LES wind speed at 75.3 m against ERA5 wind speed at 100 m, and c) histograms of 10-minute free stream LES and observed wind speed at 75.3 m during the observations-LES overlap times.

evaluating the performance of the long-term correction method of many sets of 365 consecutive days, it is insightful to consider
a graphical representation of the method and to validate the method's assumptions. To those ends, Figure 4a-b show the PDF's of wind and power production during an arbitrary year (2010) and the full run period (2010-2019), together with the estimated long-term PDF of power by applying eqn. 8. The year 2010 had below average wind speeds, which is reflected in a lower than average occurrence of rated power. The long-term correction method produces a power production PDF (black line in Fig. 4b) which visually matches the real long-term power production PDF, giving a basic first confirmation of the method's validity.
Figure 4c shows, for the years 2010 - 2019, the sensitivity of the long-term power estimate to the binning of wind in eqn. 5. In the range of about $0.5 \text{ m s}^{-1}$ to $1 \text{ m s}^{-1}$, there is a low sensitivity to the wind bin width. Outside of this range, the assumptions behind the long-term correction method break down. From now on, therefore, a wind bin width of $0.75 \text{ m s}^{-1}$ will be chosen. A simple visualization of the central assumption in eqn. 7 is shown in Figure 4d: within each wind bin starting between $3 \text{ m s}^{-1}$ and $12 \text{ m s}^{-1}$, this shows the PDF's of power production, i.e. $h_{\text{L} \mid \text{ERA}}(P, M)$, for each individual year in 2010-2019. Also, the
same PDFs but for the entire period 2010-2019 are shown ($\hat{h}_{\text{L} \mid \text{ERA}}(P, M)$). If eqn. 7 was fulfilled perfectly, the PDFs within each wind bin would coincide. Although this perfect match is not observed, their general shapes largely agree, qualititatively justifying the use of the assumption.

To make a more quantitative assessment of the validity of the assumption, the Perkins Skill Score ($S(M)$) (Perkins et al.,
2007) between $h_{\text{L} \mid \text{ERA}}(P, M)$ and $\hat{h}_{\text{L} \mid \text{ERA}}(P, M)$ can be calculated in each wind bin:

$$S(M) = \int \min(\hat{h}_{\text{L} \mid \text{ERA}}(P, M), h_{\text{L} \mid \text{ERA}}(P, M))dP, \tag{11}$$

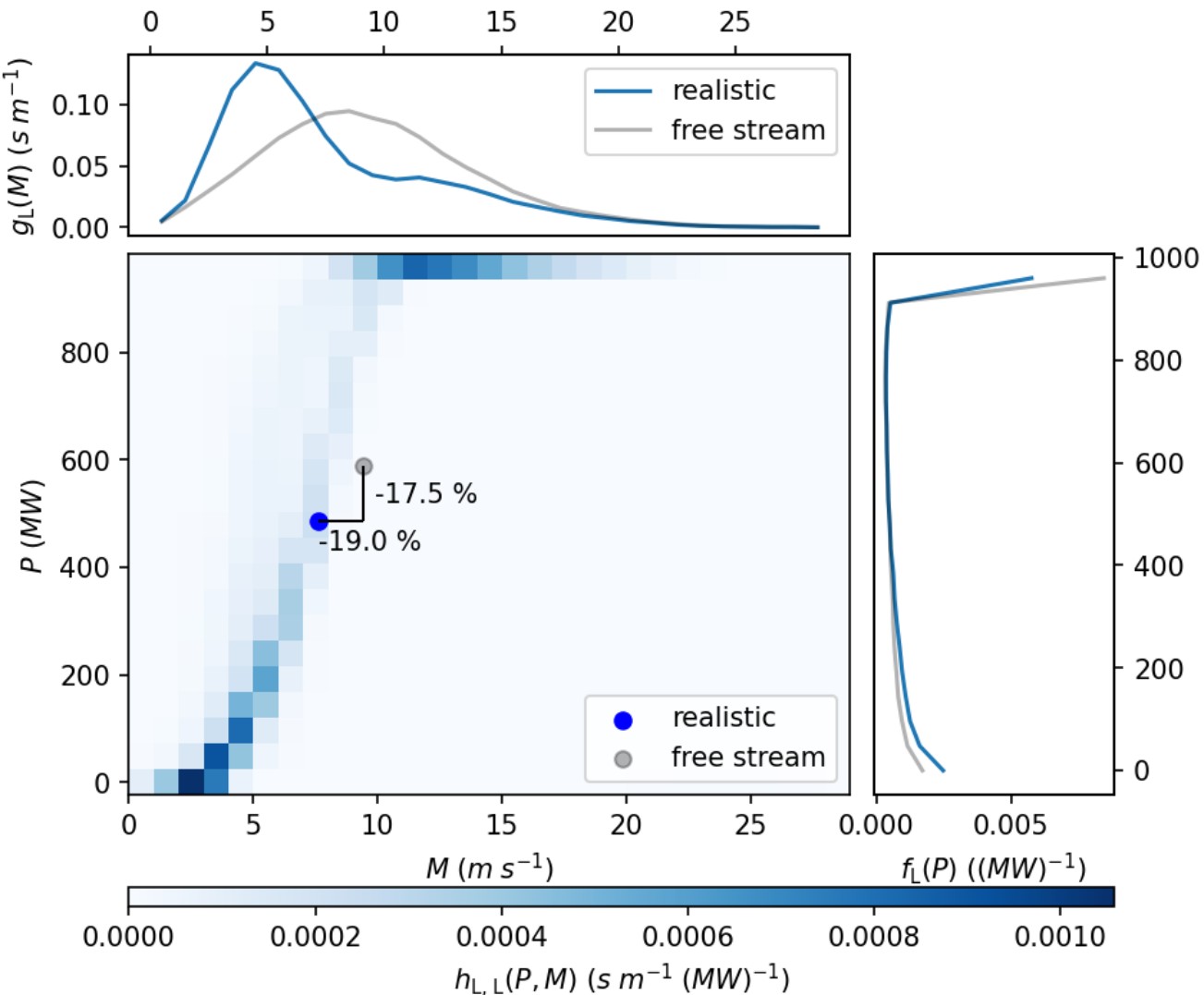

**Figure 3.** Illustration of the simulation approach. Center panel: joint PDF of LES power production and wind speed in the center of the domain at 75.3 m in the realistic run. This can also be interpreted as a probabilistic wind farm power curve. Top panel: marginal PDFs of that wind speed for the realistic and free stream runs. Right panel: marginal PDFs of power production for the realistic and free stream run. The top and right panels are the integral of the joint PDF along the vertical and horizontal axes. Dots in the center panel indicate the mean values, and percentages the decrease of those values due to the realistic inclusion of turbines in the simulation.

which measures the shared area between the long- and short-term PDFs (a 'perfect match' hence means that $S(M) = 1$). Also, the error contribution to the long-term corrected value of the power production can be calculated within each wind bin:

$$E(M) = \int ((\hat{h}_{\text{L}|\text{ERA}}(P,M) - h_{\text{L}|\text{ERA}}(P,M))\hat{g}_{\text{ERA}}(M)P dP. \qquad (12)$$

This is a useful quantity, because it reduces to the total error in the long-term corrected value when integrated over all wind bins. The last two panels of Fig. 4 show these two error metrics for 2010-2019. Three regimes are distinguishable. First, below roughly $13 \text{ m s}^{-1}$, the Perkins Skill Score is $> 0.9$; then, in the rated power regime, it is one; and above the cut-out speed it takes on a large range of values. This reflects the shape of the wind farm power curve (Fig. 4): the more spread in power at certain wind speed, the lower the lower the agreement between short- and long-term. Also, the very few occurrences of above cut-out wind speeds cause a low correspondence between short- and long term there. The error contribution per wind speed bin (Fig. 4f) corrects for this low occurrence. Note that positive and negative values of $E(M)$ can compensate. Therefore, also its absolute value averaged over the different years is shown, $\overline{|E(M)|}$. This shows that the largest absolute errors in the the long-term correction value are made in the sub-rated power regime, where the wind farm power curve shows the largest spread. It can be expected, therefore, that the long-term correction method will be less accurate for wind farms with a larger spread in power values at a certain wind speed. This can be the case for wind farms where the wake losses depend heavily on wind direction, for example due to their specific (irregular or elongated) layout.

To gain more statistical insight into the performance of the current method, it was applied on 329 series of 365 consecutive days within the 10-year LES run (each starting 10 days after the previous one); for realistic power production, free stream power production, realistic wind and free stream wind in the center of the LES domain at 75.3 m height. This allows calculation of the mean absolute error (MAE) of the long-term correction method. Figure 5 shows timeseries and their histograms of the set of consecutive days, together with their long-term corrected counterpart, and several statistical metrics. Typically, interannual variability of uncorrected variables is about 4 % of the mean, which, for power, corresponds to previously found values (Pryor et al., 2018) in a study about wind farms in the US. Applying the long-term correction removes much of that variability and produces a timeseries centered around the long-term mean value (horizontal black lines). For free stream power, realistic power, and realistic wind; the method gives MAEs of 0.36 %, 0.35 %, and 0.69 % of their long-term means, respectively. The 95th percentiles of the absolute errors are around 0.8 % (realistic and free stream power), and 1.57 % (realistic wind) of the long-term means.

### 4.3 Day selection techniques and their effect on long-term correction

When there is no need to run consecutive days in the LES (scenario 2 in the introduction), the LES run period can be chosen in order to optimize the estimation of the long-term mean power production or wind. In this section, simple methods of 'smart day selection' will be combined with the long-term correction method to get to such estimates. It should be noted that this approach of day selection is also viable without a long-term correction method. In that case, the aim is to choose days which are representative of the long-term climate (e.g. Rife et al. (2013)), and no statistical correction is applied on the simulation

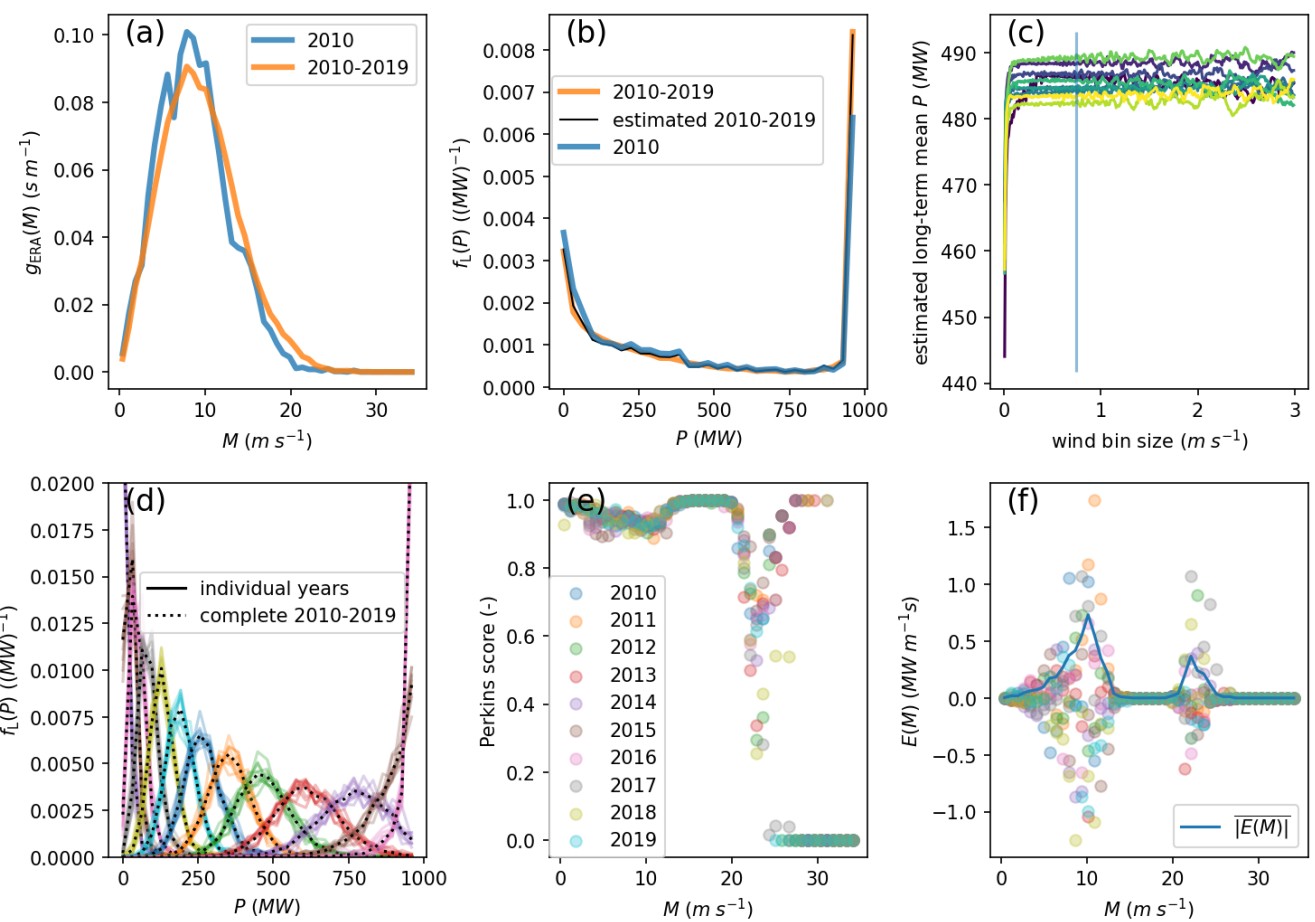

**Figure 4.** Illustration of the long-term correction method. a) ERA5 100 m wind distributions for 2010 and 2010-2019, and b) realistic LES power production distributions for 2010 and 2010-2019, and including the distribution for 2010-2019 as estimated by the long-term correction method from 2010. c) for all years 2010-2019 (increasing from blue to yellow) the long-term mean power as estimated by the long-term correction method, as a function of wind bin size. The chosen value of 0.75 m s$^{-1}$ is indicated with the vertical line. d) for wind bins starting between 3 m s$^{-1}$ and 12 m s$^{-1}$ (indicated by the different colors, increasing from left to right), the power distribution within that wind bin for the years 2010-2019 (different lines). The long-term counterparts are plotted with dotted lines. f) the Perkins Skill Score between the short- and long-term power distribution within each wind bin and for each year. f) the error contribution (and its averaged absolute value) for each wind bin and for each year (same colors as in e).

results.

The attention will now be limited to realistic power production and wind speed in the middle of the LES domain. From the 10 years of LES output, and for a range of days between 10 and 365, the day selection techniques described in Sect. 3 were applied 500 times, after which the long-term correction method was applied on the resulting samples. Resulting MAEs and

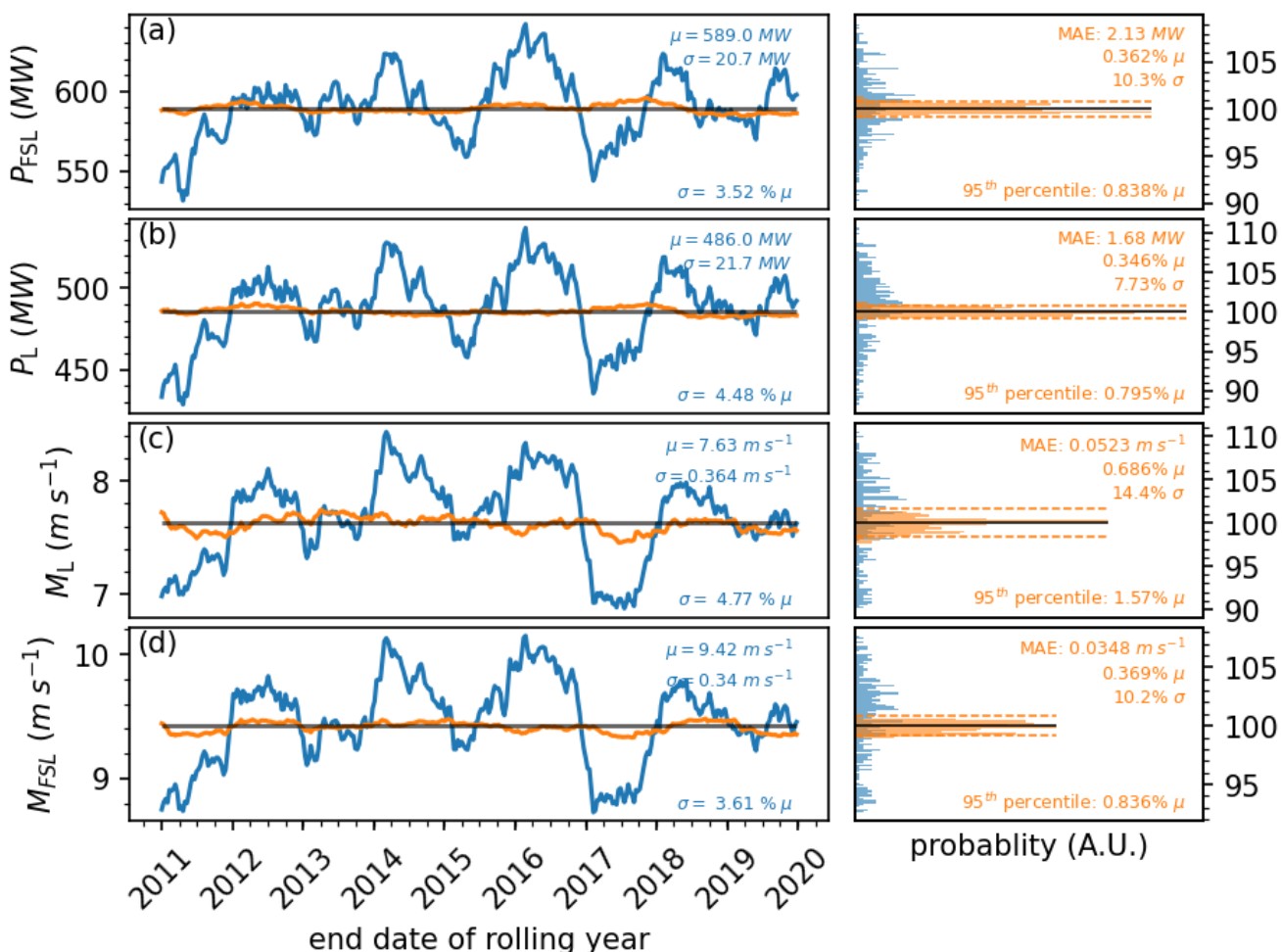

**Figure 5.** Timeseries and their histograms of long-term corrected series of 365 consecutive days. Left panels: long-term correction of (a-d) free stream power, realistic power, realistic wind, and free stream wind. Plotted values refer to the previous 365 days. Blue lines are the non-corrected values, orange the corrected values, and the horizontal grey lines show the long-term mean. Right panels show histograms of the timeseries. In the left panels, $\mu$ and $\sigma$ refer to the means and standard deviations of the uncorrected timeseries, and in the right panels, the means and 95th percentiles of the absolute errors of the long-term corrected values are given.

95th percentiles of the absolute errors are shown as a function of sample size in the right panels of Fig. 6. The left panels show the uncorrected versions, which reveal the qualitative difference between the four day selection techniques. For a sample larger than roughly 40 days, the MAE of the uncorrected power and wind decreases in the order: 'consecutive', '$k$-means', 'random', and 'ordered'. This reflects each method's representativeness of the long-term climate. By construction, $\leq 365$ consecutive days are all within the same seasonal cycle, and therefore are not representative of the long-term climate. Also, a $k$-means

algorithm applied on the two components of the horizontal wind does not represent the underlying distribution, because it

homogeneously samples all possible wind conditions, without taking into account their probability density. Only the 'ordered' method is specifically constructed to sample the wind condition weighted by their occurrence frequency, thereby making a sample which has a mean wind close the climatological value (i.e., a low MAE in the left panels of Fig. 6).

However, after applying the long-term correction method, the difference between all techniques, except for 'consecutive', has almost vanished. For the 'ordered' method, there is only a marginal improvement in the MAE compared to its uncorrected counterpart, but for the other techniques, it is substantial. For example, long-term correction of 100 random days of LES can, on average, decrease the error in estimating the long-term mean wind from 4 % to 1 %. For consecutive days, it can be decreased from 11 % to 5 %. Moreover, the previous section showed that running 365 consecutive days gives an MAE of approximately 0.35 % of the long-term mean (for power). With the 'random', 'ordered', or '$k$-means' method, this value can already be attained at approximately 200 days. Simulating more days only marginally improves the MAE, which is qualitatively consistent with the typical $1/\sqrt{N}$ behaviour of convergence of the standard error. Furthermore, if a MAE of 1 % is tolerable, fewer than 50 simulation days are required.

Even after long-term correction, the 'consecutive' method remains distinctly different from the other three, yielding considerably higher MAEs. Since it is the only method that does not select days from different years (or from different seasons, if the sample size is well below one year), this strongly suggests that choosing days from different years and seasons is the most important element in designing a day selection technique for long-term correction.

To quantify the source of the errors made in the long-term correction, Fig. 7 shows the absolute value of $E(M)$ (eqn. 12), averaged over the 500 samples, for the four day selection techniques. As observed before, the largest absolute errors are made in the sub-rated power regime (<13 m s$^{-1}$), and there is a second maximum around cut-out wind speed. This reflects the regimes where the power curve is most variable with wind speed, showing how errors in the approximation in eqn. 7 propagate into the final long-term corrected value. Like in Fig. 6, the clear decrease with sample size is visible. Also, the 'consecutive' method decreases less rapidly than the three. However, the general characteristics of the error sources are similar for all methods.

### 4.4 Including observations: correction of LES bias and climatological variability

As a final step, this section explores integration of wind observations in the long-term correction method (scenario 3). This involves performing an MCP procedure with observation data and reanalysis data, and then evaluating the integral in eqn. 10. The key difference with scenarios 1 and 2 is the use of the conditional probability $h_{\mathrm{L\,|\,FSL}}(P, M)$, i.e. the wind farm power of the realistic run, given the wind in the free stream run. In contrast to scenarios 1 and 2, therefore, integrating observations in the long-term correction procedure strictly requires the free stream LES run. The resulting long-term corrected values are harder to validate, because the simulation setup with a hypothetical wind farm precludes the use of observed wind farm power or observed disturbed wind speed.

As an illustration, a standard MCP procedure was performed on observations from 2010 (resampled to one hour), where the observed wind is linearly fitted to the reanalysis wind, in 16 wind direction sectors. Figure 8 shows scatter plots of the observed

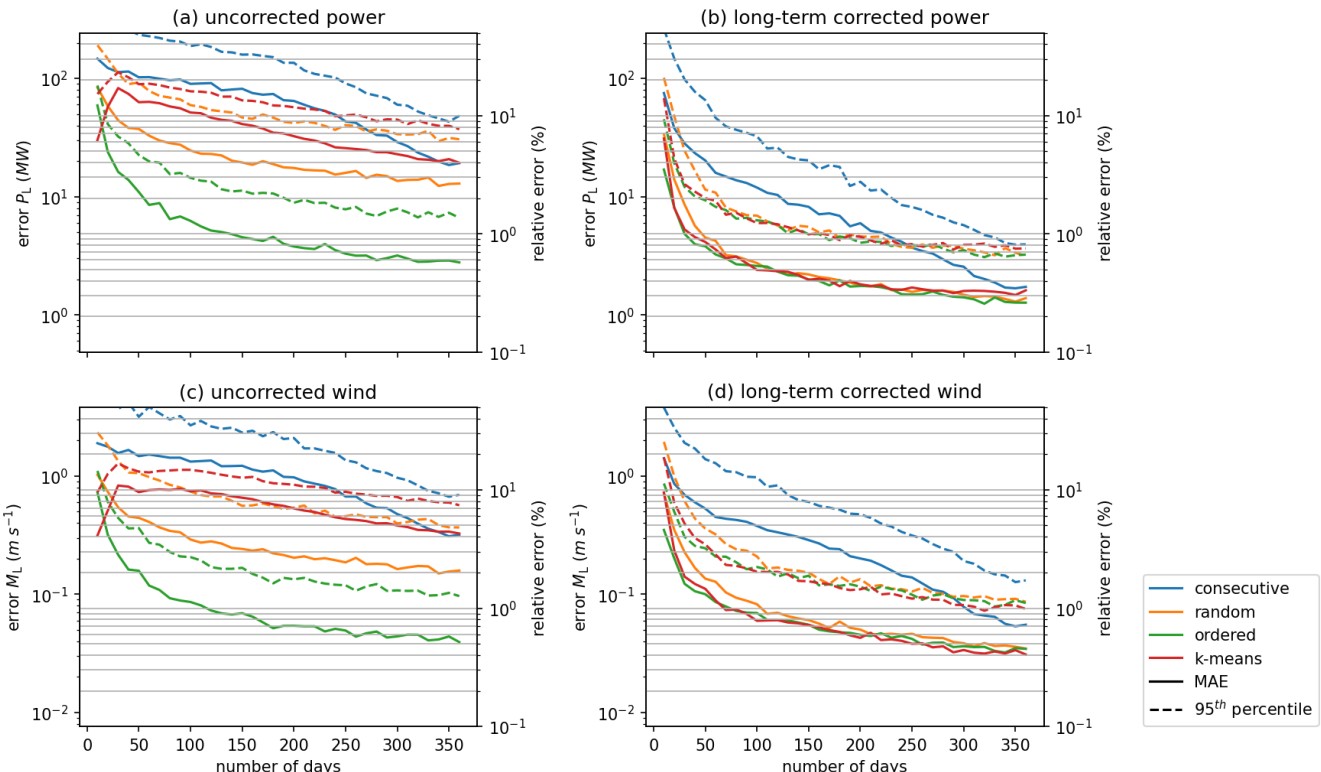

**Figure 6.** Performance of the long-term correction method for power (top) and wind (bottom) as a function of simulation period, for different day selection techniques. The left panels show the mean and 95th percentile of the absolute errors of the uncorrected samples, and the right panels show the same after applying the long-term correction method.

wind, the ERA5 wind, the free stream LES wind, and the MCP wind. By construction, the MCP wind has a zero bias with respect to the observations. The bias between the free stream LES and the MCP is very similar to the bias between the free stream LES and the observations, confirming that the MCP procedure constructs a 'quasi-observation' timeseries.

Long-term correction according to eqn. 10 was then applied for power and realistic wind (using the conditional probability
between realistic LES wind and free stream LES wind in the latter case) for 329 series of 365 consecutive days, with the previously constructed MCP wind (based on 2010). This gives values for power and wind which are corrected for both interannual variability and bias with respect to observations. It is impossible to present error metrics of the long-term corrected values of realistic wind and power, because the simulated windfarm is hypothetical. However, some useful insights can derived from timeseries of the long-term corrected values (Fig. 9). The difference between the long-term mean free stream LES wind and
the long-term mean MCP wind reflects the LES bias of about -0.36 $\mathrm{m\,s^{-1}}$. Also the difference between the long-term mean realistic LES wind and the long-term corrected wind takes on this approximate value. I.e., the long-term correction method cor-

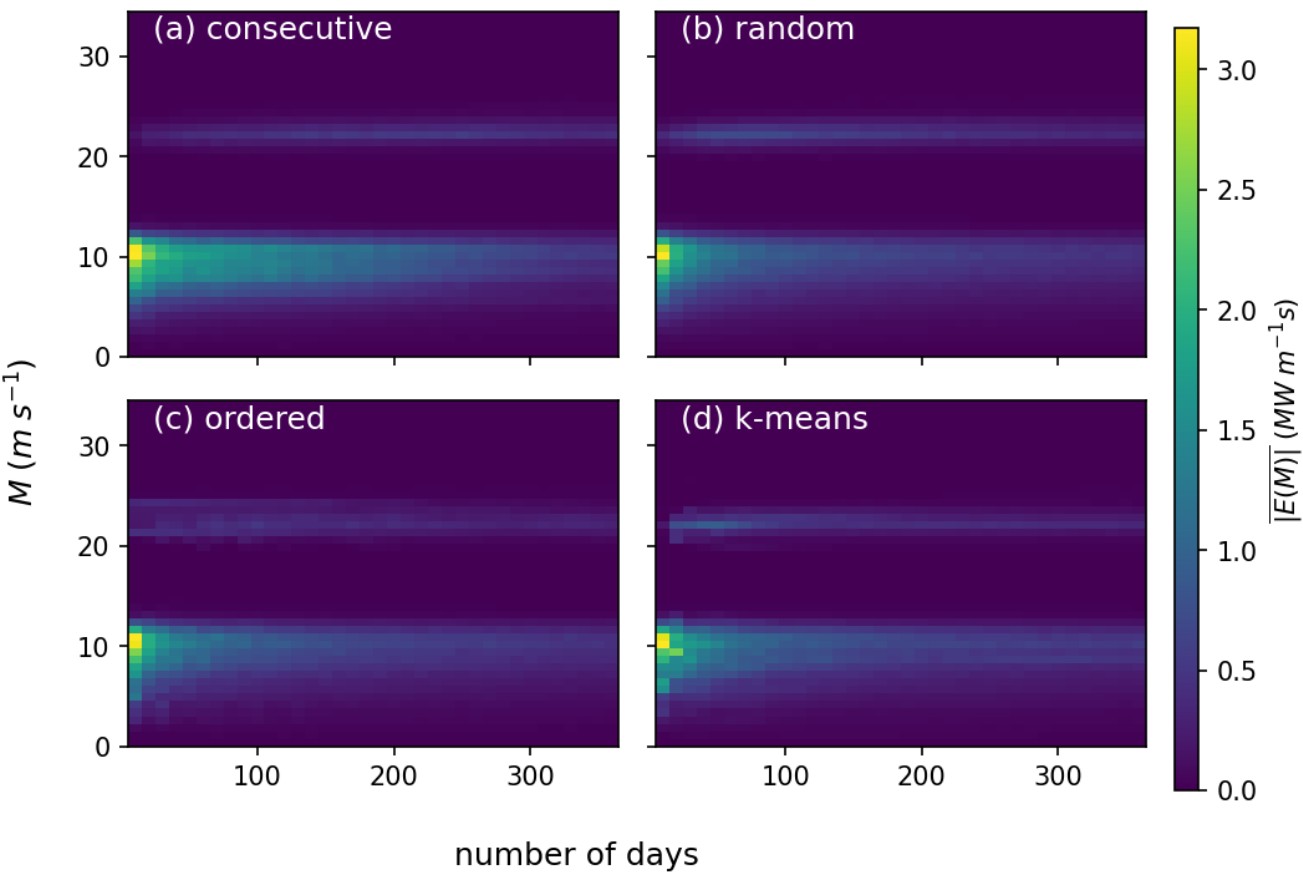

**Figure 7.** Absolute value of $E(M)$ (eqn. 12), averaged over the 500 samples, for each day selection technique.

rects a similar bias in the free stream as in the realistic case. For power, a similar pattern is observed: the long-term correction adjusts the realistic and free-stream power by roughly the same amounts ($\sim 30$ MW).

## 5  Conclusions

This work presented methods to estimate long-term mean wind and wind farm power production from shorter wind farm simulations, applied to three increasingly complex scenario's that are typical in the practice of WRA. Being applications of the definition of the conditional probability, the methods are simple, based on physical arguments, and the underlying assumptions can be verified. Furthermore, although this study uses LES as an example to verify and validate the methods, they can also be applied to other (wake) models, numerical weather prediction models, or observation data.

Data from a 10-year LES run showed that long-term correction of 365 days of pure LES wind farm power can estimate its 10-year counterpart with a MAE of around 0.35 % of the long-term mean, approximately one tenth of the interannual

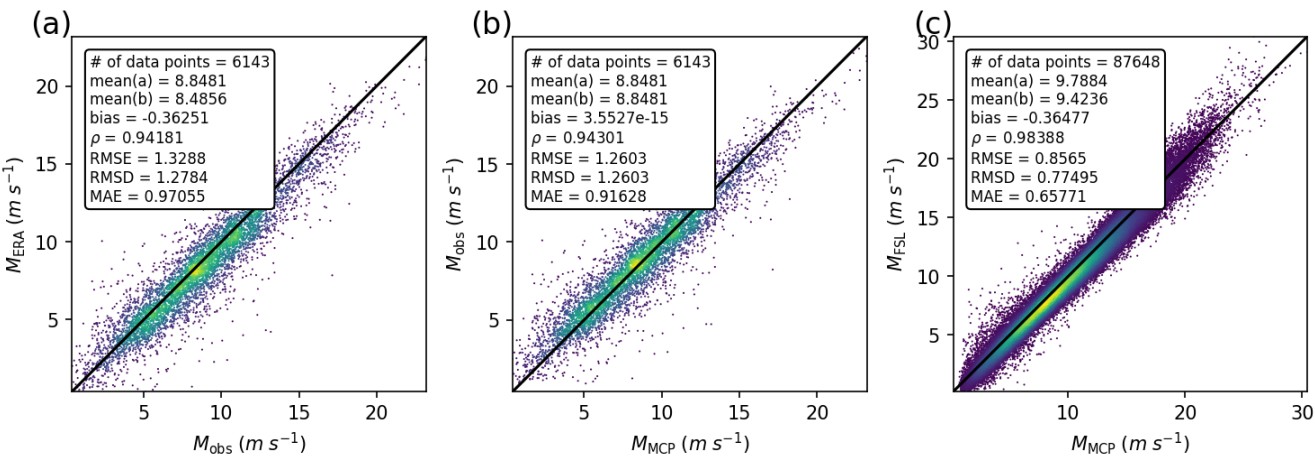

**Figure 8.** Illustration of MCP. a) 100 m ERA wind against 75.3 m observed wind during 2010, b) 75.3 m observed wind against MCP wind during 2010, anc c) free stream LES wind against MCP wind during 2010-2019.

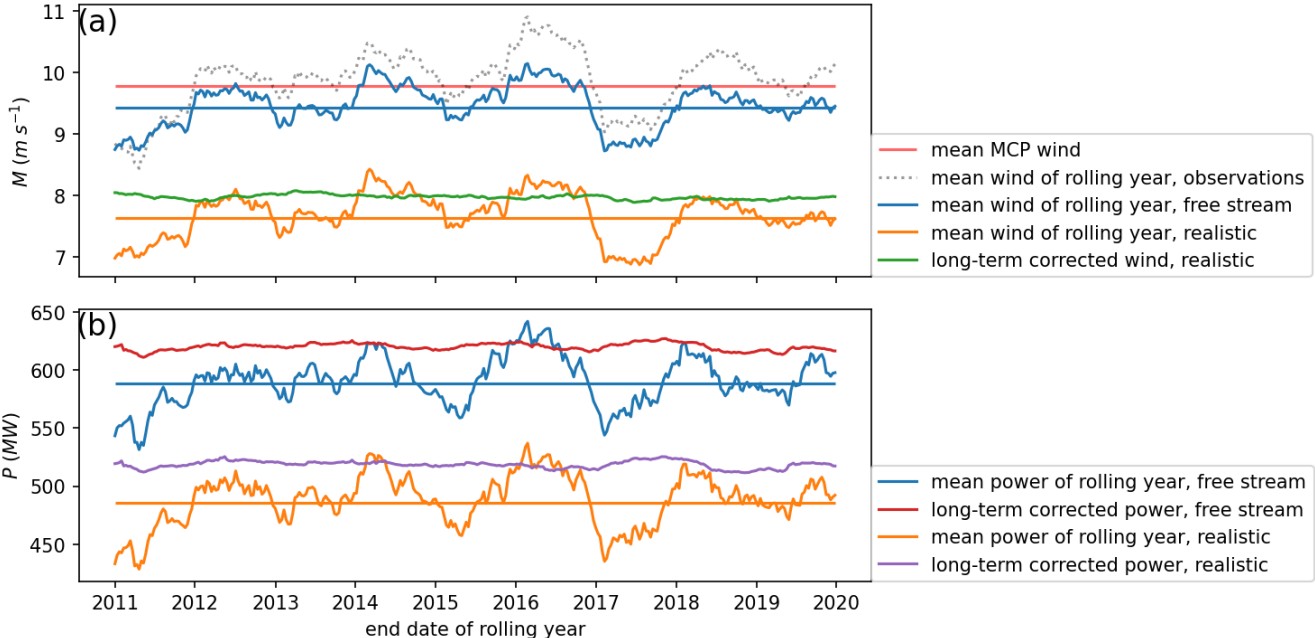

**Figure 9.** Long-term correction of 75.3 m realistic wind (a) and power (b) including observations. a) the top cluster of curves show free stream wind, including the LES, its long-term mean (horizontal blue line), and the mean MCP wind. The bottom cluster shows the realistic LES wind and long-term mean, together with its estimated long-term value (green line). b) like a), but for power production.

variability. When the run period can be freely chosen, similarly accurate estimates can be attained at periods of around 200 days. Moreover, fewer than 50 simulation days are needed to reach a MAE of 1 %. The best estimates of long-term means are achieved by choosing a sample across different years and seasons, for example with the 'ordered' method, which produces a

395 sample that is representative of the long-term climate. Nevertheless, even randomly chosen days are diverse enough to provide a good sample for long-term correction. Only for consecutive days, which are by construction during the same year, the errors are considerably larger.

Finally, it was shown that introducing observations can add value in estimating long-term means of power and wind, by correcting a possible bias in the model. This approach is more expensive, however, because an additional free stream run is

400 needed. However, such a free stream run provides additional benefit, because it allows quantification of the total wake losses of the wind farm. Furthermore, because WRA commonly is a combination of modeling and on-site observations, this method fits well within its current practices. Although validating this method against actual wind farm data was outside the scope of this research, quantifying real operational wind farm variability remains a challenge and an ongoing effort. Combining real wind farm data with methods presented in this work would therefore be a promising route for further research.

*Code and data availability.* The 10-year LES output data at one hour frequency, ERA5 wind at the same location, as well as Python code to do long-term correction according to scenario 1 are available in a Git repository, archived at Zenodo (Postema, 2024): https://zenodo.org/doi/10.5281/zenodo.11097255.

*Author contributions.* BP performed the simulations, implemented the long-term correction method, and wrote the manuscript. The concept of long-term correction was formalized by BP based on earlier work by RV and PvD. RV, PvD, PB, and HJ provided ideas, corrections, and

410 modifications. HJ and PvD are the main developers of the Aspire model.

*Competing interests.* All authors are employed at Whiffle BV, a limited liability company.

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
