# Peer review of "Estimating Long-Term Annual Energy Production from Shorter Time Series Data: Methods and Verification with a 10-Year Large-Eddy Simulation of a Large Offshore Wind Farm"

_Wind Energy Science, 2024_

## Referee Comment (RC1)

Review of manuscript "Estimating Long-term annual energy production of large offshore wind farm from large-eddy simulations: methods and validation with a 10-year simulation."

The authors present three different approaches to represent long-term effects (defined in the order of 10-year averages) in a hypothetical wind farm in the North Sea by using large-eddy simulations of a period smaller than that of a full calendar year. Their approach and results are interesting; however, I would only recommend their work for publication after they carefully address the following:

1) In section 2.1 the authors layout the foundation of their approach. According to the authors, their approach involves the application of Bayes' theorem to the continuous random variable P which is the power production of the entire wind farm, and the continuous random variable M which is the wind speed at a nearby location. To this end, the authors define the conditional densities in equations (1) and (2) however they do not explain how Bayes' rule is applied. What appears to be happening is the conditional densities are integrated to obtain the probability density for the wind farm power production obtained by means of LES, $f_L(P)$. This is indeed based on Bayes' theorem, but further explanation is needed. If we start from Bayes' theorem we have:

$$h_{L \mid ERA}(P, M) = \frac{h_{ERA\mid L}(M, \ P) \, f_L(P)}{g_{ERA}(M)},$$

where $h_{L \mid ERA}(P, M)$ is the conditional probability of the power predicted correctly by LES given that the ERA5 reanalysis data provides true values for the wind speed, equals to the likelihood of the wind speed being calculated correctly by ERA5 given that power is calculated correctly by LES and multiplied with the prior probability density, $f_L(P)$, and the marginal probability density, $g_{ERA}(M)$. What the authors do not mention is their main premise which is that the likelihood of the wind speed being calculated correctly by ERA5 given that power is calculated correctly by LES is equal to 1. This allows them to integrate over the wind speed M to obtain

$$\int h_{L \mid ERA}(P, M) g_{ERA}(M) dM = \int f_L(P) \, dM = f_L(P)$$

During integration they also use the fact that $f_L(P)$ is independent of the wind speed (and that $\int dM = 1$). To apply Bayes' theorem the authors, need to also consider that $f_L(P)$ and $g_{ERA}(M)$ to be independent probability densities, which I guess is self-evident by the fact that the two distributions have been synthesized from different datasets. This is also something the authors need to emphasize.

2) The second assumption they make is that the conditional probability between power and wind calculated for 1 year approximates the long-term counterpart. This allows the authors to calculate the long-term probability density of power $\hat{f}_L(P)$, by

only using information from the long-term wind speed probability density, $\hat{g}_{ERA}(M)$ which can be easily obtained from the ERA5 record, and the conditional probability calculated from the down-selected days, $h_{L\,|\,ERA}(P,M)$. This assumption is attempted to be validated in section 4.2, and more specifically in figure 4d, however

    a. Data are shown only for scenario 1 (full-year simulation)

    b. The phrase "their general shapes largely agree" in line 248-249 cannot be used instead of a quantitative metric.

My recommendation to the authors would be to use a rigorous metric such as the Perkins Skill Score (PSS) or a goodness of fit test, such as the Kolmogorov-Smirnov test, to measure how well the conditional probability densities, $h_{L\,|\,ERA}(P,M)$ and $\hat{h}_{L\,|\,ERA}(P,M)$ agree with each other. Rigorously quantifying the matching between the 1-year and long-term conditional probabilities, will provide more value to the study and increase its overall impact.

3) LES resolution may not be sufficient. While I fully agree that the authors have provided results from a state-of-the-art, meso-microscale coupled model and have therefore been pushing the limits of wind farm modeling (including resolution), the statement "...*Although this can be considered a coarse resolution, Baas et al. 2023 showed that refining to 60 m has a relatively small effect on total aerodynamic losses of a 770 MW wind farm...*" is problematic. The reasons are the following:

    a. The two studies consider different size wind farms 960MW versus 770MW and different array densities 7.2MW/km2 versus 10MW/km2. This may result in a different number of nodes used to cover the turbine spacing, so it is not really a direct comparison.

    b. Wakes remain unresolved when using either a 120m or a 60m resolution and therefore the small change in power losses should not be used justify accuracy particularly when lacking validation.

    c. The authors in Baas et al 2023, provide a much better reason for why a resolution of 120m is selected: "*This choice results from a trade-off between computational cost and accuracy and has been tested extensively in an operational setting*". I fully agree with this statement. Such studies have been pushing the state of the art of offshore wind farm modelling and they must not be judged based on previous LES studies that have only considered canonical ABL cases (or what I call turbulence in a box). I suggest the authors re-phrase this part of the paper to provide a similar statement.

Overall, the paper presents a novel an interesting approach (based on Bayes' rule) to correct for long-term effects, but the structure of the paper is not clear, and it requires additional effort by the reader. In addition, the presentation of the results needs also to be improved to allow the authors to better highlight their key findings.

---

## Author Comment (AC2)

We thank the reviewer for their comments on the manuscript. Although the discussion has not ended yet, we will already reply on this review here, and incorporate the second one later. The reviewer has raised three very valid points, which we plan to address in the following way:

1. We agree that our explanation of the use of Bayes' theorem (in section 2.1) is lacking. In fact, in the long-term correction method, we only use the definition of the conditional probability density, from which Bayes' theorem is a direct result. For example, for continuous random variables X and Y, the definitions of the conditional probability densities are:

$$f_{X|Y}(X,Y) = \frac{f_{X,Y}(X,Y)}{f_Y(Y)}, \text{ and } f_{Y|X}(X,Y) = \frac{f_{X,Y}(X,Y)}{f_X(X)}, \tag{1}$$

and therefore:

$$f_{X|Y}(X,Y) = \frac{f_{Y|X}(X,Y)f_X(X)}{f_Y(Y)}. \tag{2}$$

Eqn. 2 is Bayes' theorem, whereas in the manuscript, we only used eqn. 1, in the following form:

$$h_{\text{L | ERA}}(P,M) = \frac{h_{\text{L, ERA}}(P,M)}{g_{\text{ERA}}(M)}, \tag{3}$$

from which the remainder of the derivation follows.

Therefore, we should not have written that we use Bayes' theorem, but that we apply the definition of the conditional probability density (from which Bayes' rule is the direct result). We have adapted this in the manuscript, and added further explanation.

2. A more rigorous quantification of the match between short- and long-term conditional probability densities is indeed a good idea. We therefore added (for scenario 1) the Perkins Skill Score as a function of wind bin ($S(M)$) in the manuscript's Fig. 4:

$$S(M) = \int \min(\hat{h}_{\text{L | ERA}}(P,M), \ h_{\text{L | ERA}}(P,M))dP. \tag{4}$$

The new version of the manuscript's Fig. 4 is shown here in Fig. 1. Also, in a similar way, the error in the long-term corrected value can be split up per wind bin, thereby quantifying the final effect of the degree to which the approximation $h_{\text{L | ERA}}(P,M) \approx \hat{h}_{\text{L | ERA}}(P,M)$ holds:

$$E(M) = \int ((\hat{h}_{\text{L | ERA}}(P,M) - h_{\text{L | ERA}}(P,M))\hat{g}_{\text{ERA}}(M)PdP. \tag{5}$$

Because it takes into account the mean power production value in each wind bin, and the occurrence frequency of each wind bin, $E(M)$ integrates to the

[Figure]

Figure 1: Updated version of Fig. 4, including Perkins Skill Score and $E(M)$.

total final error. For our specific purposes, this makes it a useful metric. Therefore, we intend to add a figure that shows, for each day selection technique and sample size, the absolute value of $E(M)$ (here shown in Fig. 2).

3. Our relatively coarse resolution indeed is a between accuracy and computational cost. Apart from adding a resolution sensitivity study (see our reply on the first community comment), we will emphasize this more in the manuscript.

[Figure]

Figure 2: Intended new figure showing the absolute value of $E(M)$ per wind bin and per sample size, averaged over the 500 different samples. This quantifies the source of the error caused by the imperfect match between the short- and long-term conditional probability densities.

---

## Author Comment (AC3)

Dear editor,

We thank the reviewer for their comments on the manuscript. The reviewer raises a number of concerns related to the resolution of our simulations. We will address the specific comments below, but we first would like to make a few more general points.

While there may be a justified discussion about the resolution of the model, this is beside the main point of the paper. We could have presented exactly the same method for long-term correction based on 10 m LES runs, a mesoscale simulation or even another model, as long as a self-consistent dataset of 10 years is available. The contribution of our paper is not to present or validate our LES model. It is the statistical correction method. Because LES is gaining popularity in WRA, however, we think that demonstrating the statistical correction method using LES output increases the relevance of the method. Then, being able to present two 10-year simulations driven by real-weather data outweighs the drawbacks of simulating on a relatively coarse resolution.

Secondly, since the resolution was also a concern of the first community comment, we have provided a document with supplementary analyses (at the bottom of this document), which will also serve as supplementary material to the manuscript. We believe this complement addresses a lot of the concerns raised by the reviewer.

Thirdly, there seems to be some confusion on the origin of the ASPIRE model, and its relation to GRASP (GPU-Resident Atmospheric Simulation Platform). ASPIRE is the name for our entire modelling suite; which encompasses the LES core, called GRASP, as well as various other components (e.g. the turbine calculations, the land-surface calculations). Previously, work using our model often referred to it as GRASP. Here, we choose to refer to the entire modelling suite. We indeed understand the confusion, and will clarify the manuscript.

The reviewer's comments are in italics, our responses in normal font.

General remarks

*While the numerical study presented in the manuscript is unique because it uses a long-term (10-year) LES, the authors did not present a convincing motivation and justification for their numerical study that uses coarse LESs to estimate offshore wind speed and simulate wind farm power production by representing operating wind turbines with an actuator disk model. It is not clear that a coarse LES can deliver any benefit in estimating hub height or rotor equivalent wind speed compared to significantly less costly mesoscale simulations. The LES at resolution of 120 m does not improve wind speed prediction compared to mesoscale simulations (> 1 km) or even ERA5 reanalysis as shown in the manuscript. Coarse LES does not resolve turbulent eddies in the inertial range considering: offshore conditions, effective resolution, and the need for inflow turbulence development not described in the manuscript.*

A motivation and justification for the use of coarse LES for wind energy purposes is that it is a computationally feasible approach for estimating AEP or

the wind climate in a certain location. Current approaches to estimating wind climate or AEP for offshore wind farms include engineering wake/flow models, mesoscale models, RANS and LES. Each of the methods has its advantages and limitations. While many LES studies presented in the scientific literature use a much higher resolution, they are then usually limited to a small number of cases. To estimate AEP, however, either a large set of flow cases or a large set of days needs to be modeled, to include all prevailing wind conditions that the wind farm will encounter. So coarse resolution LES for wind resource AEP assessment provides a viable alternative and is now used routinely in the wind energy sector. We have added a number of extra references to make this point clearer. Specifically, the references to work with ASPIRE, including validation, for wind energy now are: the work by Williams et al. (2024), Oldbaum (2019), Baas et al. (2023) (wind farm modelling); Schepers et al. (2021), Taschner et al. (2023) (turbine physics and loads); Gilbert et al. (2020), Alonzo et al. (2022) (wind forecasting); and Kantharaju et al. (2023), Storey and Rauffus (2024) (wind climate modelling).

*The study presents a Bayesian downscaling approach that combines LES and reanalysis output. While this approach is interesting and the use of LES can be considered as something that has not been explored before, similar methodologies have been used extensively for regional climate downscaling (e.g., Holthuijzen et al., 2021, J. App. Meteorolo. Climat. and references therein). The authors did not provide any references to research that preceded their study, so based on the way the approach is presented, it would seem that this approach is completely novel. The novel part is exploring three scenarios for long-term correction, and this is possibly the most relevant contribution.*

We thank the reviewer for pointing this out and for providing a reference with similar correction methods. At the same time, we feel that our method is different in a number of ways from e.g. the methods described in Holthuijzen. Namely, rather than refining a coarser simulation (i.e. downscaling), our method estimates the long-term probability distribution of a variable obtained from a short-term fine-scale simulation. Nevertheless, we do agree that it is useful to position our method compared to other methods in the introduction, so we have added the following to the manuscript:

"Correcting or downscaling weather- or climate model data to better match observed reality is a widespread practice in environmental science. In this field, two main types can be distinguished (e.g. Ekström et al., 2015; Holthuijzen et al., 2021): dynamical downscaling, in which a more accurate model is used to refine results from an often coarser model; and statistical downscaling, in which statistical relationships between the modelled and observed variable are used to correct the modelled variable. These methods generally produce corrected timeseries, but can include spatial dimensions as well (Holthuijzen et al., 2021). The current study uses aspects of both the dynamical and statistical type. Firstly, the data to be corrected itself come from an LES downscaling of ERA5. Secondly, and this is the core message of the study, the probability distributions are then corrected for the difference in weather conditions between

the simulation period and the long-term climate. This is done by estimating the conditional probability density between the variable of interest and ERA5 wind from the short simulation, and then using this relationship to modify the probability density of the variable of interest to represent the long term. In this sense, the long-term correction method is most related to statistical downscaling, but instead of correcting for a model deficiency, the long-term correction method corrects for non-representative simulation periods."

*The study uses an actuator disk model that is not appropriate for the coarse LES resolution used in the study, i.e. 120 m in horizontal and 30 m in vertical taking, with the effective resolution that is even coarser. For example, Calaf et al. (2010, also Meyers and Meneveau, 2010, cited in the paper) resolved the rotor with 10 vertical grid cells. Baas et al. (2023) conducted a sensitivity study with grid cell size of 120 m and 60 m and similar vertical grid as present study and argued that the 120 m grid cell size is adequate, however, their estimates of aerodynamic losses were about 20% larger with coarser resolution – this is significant. In addition, the actuator disk model was implemented in the ASPIRE LES model, a version of a commercial, GPU-based model, which is not publicly available. It is not clear how ASPIRE differs from other versions of the GPU model (e.g., GRASP used in Baas et al., 2023, study) and if ASPIRE was validated in any way.*

The use of an actuator disk model at this resolution is indeed a relevant topic of discussion, and to provide more context, the supplementary material shows how wind farm dynamics depend on resolution. Indeed, like in Baas et al. (2023), we find that a coarser resolution results in higher losses. This is mainly due to opposing responses of power production to resolution in the free-stream and realistic runs. However, for the purposes of this study, the response of the actuator disc model to resolution is not important: presenting and validating the long-term correction methods is the main message. In order to validate them with 10 years of LES data, therefore, a trade-off between accuracy and computational cost had to be made. Given the computational resources, one has to balance resolution, horizontal domain size, and vertical domain size. The latter two are important for representing global blockage and gravity waves (respectively). Making this trade-off resulted in the choice of a 120 m resolution, horizontal domain size of 10 km, and vertical domain size of 3 km.

As for the last point about validation of ASPIRE, please see the references in our reply to the first general remark.

*Although, to achieve greater realism and therefore relevance, the numerical study includes coupling between coarse reanalysis and LES simulated is a hypothetical wind farm which means that the results of the study cannot be validated. The use of a hypothetical wind farm is of a very limited value considering that it is now possible to obtain wind power production data for existing wind farms in the North Sea (see, e.g., https://rave-offshore.de/en/data.html).*

We agree that the validation and quantification of uncertainty of AEP calculations based on LES with actuator disk is important and deserves more

research. However, we feel that validation would be widening the scope of the paper too much. Furthermore, applying the long-term correction methods to actual wind power production data could be a relevant route for further research. Here however, to present and validate the methods, we chose to use LES data, because it fits well with current WRA practices. In WRA, studying hypothetical wind farms is a very wide-spread practice.

Specific remarks

*Abstract – The abstract is confusing and poorly written, e.g., no motivation for the use of LES is provided.*

We will rephrase the abstract to provide more motivation for the use of LES.

*Line 28 – It is stated that "'real- weather' LES has been demonstrated to be viable... to explicitly model the interactions between wind farms and the atmosphere (Baas et al., 2023)" however the cited study by Bass et al. (2023) does not include a validation that would demonstrate viability.*

and

*Line 68 – ASPIRE model is used in the study and a reference is made to the Dutch Atmospheric Large Eddy Simulation (DALES) model and the GPU version, however, there are no previous references to the ASPIRE model and its validation.*

We have clarified that we have adopted ASPIRE as the new name to refer to our entire modelling suite, and have added a number of extra references about its origins, use, and validation (see the reply to the first general comment).

*Line 119 – It is stated that "In the practice of LES modelling, it is often found that the wind speed displays a mean bias of O(0.1 m/s)..." but no reference is provided for this statement.*

This statement is indeed not well-supported, it stems from our own practical experience of the model. We have added two references that support it (Storey and Rauffus (2024) and Kantharaju et al. (2023)).

*Line 121 – The sentence: "Such a free stream run has no turbines included, or it has turbines that exert no force on the flow, and therefore leave it undisturbed." can be removed since is nonsensical. Turbines that exert no force on the flow are not turbines, since even turbines at rest exert force on the flow.*

We respectfully disagree that this is a nonsensical statement. It is common in wake studies to calculate what is sometimes called 'gross power', by using the undistributed wind speeds as if no turbines were present. Of course this is a theoretical concept, but nonetheless useful and widely accepted.

*Line 131 - It is stated that "... because of explicit representation of all fluid dynamical... effects..." First, it is not true that all fluid dynamical effects are represented since the rotation of the turbine is not accounted for, and second, at the coarse resolution used in the study it cannot be said that the fluid dynamical effects are represented well.*

We agree that not all fluid dynamics is explicitly represented, and will change this in the text. However, we do believe that the key processes that govern real-weather boundary layer-flow and wind park power production are represented adequately. Again, we refer to the supplementary material for a more thorough discussion of this.

*Line 160 – How is ASPIRE different from or related to GRASP. Has it been validated?*

See our previous remarks about ASPIRE and GRASP. Concerning validation, we have added a number of studies (see reply to the first general comment).

*Line 164 – Given the resolution, it is not clear how turbulence develops on the inner LES domain.*

More information on turbulence generation is indeed a valid request, since it is an essential part of real-weather LES modelling. Turbulence is generated by adding fluctuations to the inflow conditions from the mesoscale simulations. These fluctuations are themselves produced by a smaller periodic precursor LES that is driven by the mesocale field at the LES boundaries. Hence, these added fluctuations are consistent with atmospheric conditions (Storey and Rauffus, 2024). We have added this information to the manuscript.

*Line 165 – It is stated that "...the core LES domain is nested in a coarser mesoscale-type simulation with a resolution of 1.5 km,..." and "This coarser simulation has the same model formulation as the LES...," however, it is not clear what is the exact model formulation. It is only mentioned that the turbulence parameterization is different, but not how is it different, what other parameterizations were used: radiative transfer, microphysics, land surface, etc. For the study to be reproducible this information should be provided.*

We agree that we have been unclear about the other parametrizations, and have adapted this in the manuscript.

*Line 185 – If turbines do not exert any force they are not included in the simulation, so there is no need to mention turbine.*

See our earlier remark about the use of free-stream turbines.

*Line 221 (also Figure 2) – It is not clear what is the benefit of LES offshore when it results in larger errors and turbulence is not resolved with 120 m grid cell size.*

See our introductory remark.

*Line 230 – As shown by Baas et al. (2023) a coarse LES result in overestimation of aerodynamic losses.*

This is indeed a known effect, which is also demonstrated in the supplementary material. However, it does not affect the core message of the paper: presenting and validating the long-term correction methods with a 10-year run.

**References**

Alonzo, B., Cassas, M., Raynaud, L., Verzijlbergh, R., Houf, D., Baas, P., and Dsouza, B.: Smart4RES: Report on improved NWP with higher spatial and temporal resolution, `https://www.smart4res.eu/wp-content/uploads/2023/01/Smart4RES_Deliverable_D2.2.pdf` (last accessed 9-8-2024), 2022.

Baas, P., Verzijlbergh, R., Dorp, P. V., and Jonker, H.: Investigating energy production and wake losses of multi-gigawatt offshore wind farms with atmospheric large-eddy simulation, Wind Energy Science, 8, 787–805, https://doi.org/10.5194/wes-8-787-2023, 2023.

Ekström, M., Grose, M. R., and Whetton, P. H.: An appraisal of downscaling methods used in climate change research, Wiley Interdisciplinary Reviews: Climate Change, 6, 301–319, https://doi.org/10.1002/wcc.339, 2015.

Gilbert, C., Messner, J. W., Pinson, P., Trombe, P. J., Verzijlbergh, R., van Dorp, P., and Jonker, H.: Statistical post-processing of turbulence-resolving weather forecasts for offshore wind power forecasting, Wind Energy, 23, 884–897, https://doi.org/10.1002/we.2456, 2020.

Holthuijzen, M. F., Beckage, B., Clemins, P. J., Higdon, D., and Winter, J. M.: Constructing High-Resolution, Bias-Corrected Climate Products: A Comparison of Methods, Journal of Applied Meteorology and Climate, 60, 455–475, https://doi.org/10.1175/JAMC-D-20, 2021.

Kantharaju, J., Storey, R., Julian, A., Delaunay, F., and Michaud, D.: Wind resource modelling of entire sites using Large Eddy Simulation, in: Journal of Physics: Conference Series, vol. 2507, Institute of Physics, ISSN 17426596, https://doi.org/10.1088/1742-6596/2507/1/012015, 2023.

Oldbaum: Wind Resource Assessment Hollandse Kust (noord) Wind Farm Zone, `https://offshorewind.rvo.nl/files/view/9717fb65-79ab-4966-92e2-a73c856c18c9/hkn_20191022_oldbaum_wra-oct19-f.pdf` (last accessed 9-8-2024), 2019.

Schepers, G., Dorp, P. V., Verzijlbergh, R., Baas, P., and Jonker, H.: Aeroelastic loads on a 10 MW turbine exposed to extreme events selected from a year-long large-eddy simulation over the North Sea, Wind Energy Science, 6, 983–996, https://doi.org/10.5194/wes-6-983-2021, 2021.

Storey, R. and Rauffus, R.: Mesoscale-coupled Large Eddy Simulation for Wind Resource Assessment, in: Journal of Physics: Conference Series, vol. 2767, Institute of Physics, ISSN 17426596, https://doi.org/10.1088/1742-6596/2767/5/052040, 2024.

Taschner, E., Folkersma, M., Martínez-Tossas, L. A., Verzijlbergh, R., and van Wingerden, J. W.: A new coupling of a GPU-resident large-eddy simulation

code with a multiphysics wind turbine simulation tool, in: Wind Energy, John Wiley and Sons Ltd, ISSN 10991824, https://doi.org/10.1002/we.2844, 2023.

Williams, S., Dubreuil-B, C., and Seim, K. S.: Multi-fidelity wake model validation at the Arkona offshore wind farm, conference session at Wind Europe Technology Workshop, `https://windeurope.org/tech2024/programme/proceedings/sessions/the-impact-of-resource-on-performance/` (last accessed 9-6-2024), 2024.

**Supplementary Material: A Resolution Study on Large Eddy Simulation of the Horns Rev Wind Farm**

**August 2024**

**1 Introduction**

In this supplementary study, we present additional LES runs of the Horns Rev offshore wind farm, at varying resolutions. The goal is show the effect of LES resolution on wind farm power production, and profiles of wind, turbulence kinetic intensity, and the fraction of subgrid-scale and resolved turbulence kinetic energy. By performing both realistic runs (with active turbines) and free-stream runs (without active turbines), we can investigate the representation of wind-farm dynamics and boundary layer meteorology separately.

**2 Methods**

Like in the main text, the ASPIRE model is employed; with an LES with open boundaries nested in a coarser mesoscale-type simulation that does not resolve turbulence. The domains are centered at the Horns Rev wind farm (55.49°N, 7.48°E), and the run period consists of 10 days[1]. These 10 days were chosen by applying a $k$-means algorithm on the daily mean 100 m horizontal components of the ERA5 wind speed at that location, during 2015—2016.

The domain size of the coarser simulation is 256 km by 256 km, with a horizontal resolution of 2 km, and the 64 vertical levels start with a spacing of 40 m and stretch exponentially to the domain top at 8 km. Figure 2 shows the simulation setup. The LES domains vary in resolution and number of grid points as shown in Table 2, but all have a horizontal size of 15360 m by 15360 m. This ensures that all LES runs have same lateral inflow conditions. Above 200 m, the LES runs use an exponentially stretched vertical grid, which varies for the different resolutions. For each resolution, both a realistic (active turbines) and free-stream run (turbines leave the flow undisturbed) are performed.

The simulated wind farm is the well-studied Horns Rev offshore wind farm, which has 80 turbines of the V80-2.0MW type, and a total rated power of 160 MW. The details of the wind farm modelling are discussed in the main text and in Baas et al. (2023).

**3 Results**

The results from free-stream runs, providing insight into the effects of LES resolution on lower boundary-layer dynamics, are discussed in subsection 3.1. Then, subsection 3.2 presents the effects of LES resolution on wind farm dynamics. Here, results from both the free-stream and realistic runs are used to quantify the aerodynamic effects of the wind farm.

**3.1 Effects of resolution on boundary layer dynamics**

For all different resolutions, Fig. 3.1 presents vertical profiles of horizontal wind ($M$), its 10-minute standard deviation ($\sigma_M$), turbulence intensity (TI $= \sigma_M/M$), turbulence kinetic energy (tke) as resolved by the model, and its subgrid-scale (sgs) value, and finally the ratio of resolved to subgrid-scale tke.
* * *
[1]2015-01-13, 2015-02-14, 2015-06-11, 2015-08-31, 2015-09-01, 2015-10-29, 2016-02-21, 2016-07-26, 2016-10-18, 2016-12-21.

[Figure]

Figure 1: The simulation setup. a) the LES (small orange square) is nested in a coarser simulation (large orange square) on the North Sea. b) The LES domain with the wind farm layout (dots).

All these quantities show their expected mean vertical profile in the (lower) boundary layer. In general, there are small differences between the resolution, which decrease with height. For mean horizontal wind, the differences between the resolutions become negligible above approximately 100 m. Below that, they are of the order of 0.1 m s$^{-1}$. In the rotor layer, $\sigma_M$ varies between approximately 0.5 m s$^{-1}$ and 0.7 m s$^{-1}$. Combined with a wind speed between 9 m s$^{-1}$ and 10 m s$^{-1}$, this gives a mean TI of about 0.04 to 0.07. For both these quantities, the variation with height is larger than the variation between the resolutions.

The total tke in the rotor layer varies between approximately 0.35 m$^2$ s$^{-2}$ and 0.55 m$^2$ s$^{-2}$. Of this total tke, more than 92% is resolved for all resolutions.

The presented turbulence quantities show a resolution-dependence that might be counter-intuitive. Namely, they often increase with decreasing resolution. This is a result of the choice of subgrid turbulence parameterization used in this study. The ASPIRE model in its current set-up uses an anisotropic minimum dissipation subgrid turbulence parametrization. In this parametrization, the subgrid scheme becomes less diffusive (i.e.: there will be a more turbulent flow) when the aspect ratio of the grid boxes is higher. The simulations with a coarser resolution have a higher aspect ratio, and therefore a less active subgrid scheme, and hence a higher resolved tke.

These profiles presented until now are averaged over the whole simulation period, which might obscure a mean diurnal cycle. However, Fig. 3.1 shows that, as is common over sea, there is only a weak mean diurnal cycle in wind and turbulence quantities. The peak in hourly mean tke at 21:00 can be traced back to one strong wind ramp event on August 31st 2015, related to the passage of the core of a low pressure system very close to the wind farm. This wind ramp caused a roughly 4 m s$^{-1}$ spike in wind speed (at hub height) over the course of a few minutes. Since this variability on the sub-10-minute scale, it causes an strong increase in the 10-minute tke.

**3.2   Effects of resolution on wind farm dynamics**

The inclusion of wind turbines in the simulations allows us to study the effect of LES resolution on wind farm dynamics, and in particular, power production. Also, by comparing this realistic run to the free stream

Table 1: Domain setup of the LES runs. dx and dy are the horizontal grid spacing, dz the vertical grid spacing below 200 m, Lz is the domain height, and Nx and Ny are the number of grid points in the horizontal direction. Lz and Nx and Ny are chosen such that the horizontal domain size is 15360 m for all runs.

| simulation id | realistic / free stream | dx, dy (m) | dz (m) (surface) | Lz (m) | Nx, Ny |
|---|---|---|---|---|---|
| 000 | free stream | 120 | 30 | 2515 | 128 |
| 001 | free stream | 80 | 20 | 2547 | 192 |
| 002 | free stream | 40 | 20 | 2547 | 384 |
| 003 | free stream | 20 | 20 | 2547 | 768 |
| 010 | realistic | 120 | 30 | 2515 | 128 |
| 011 | realistic | 80 | 20 | 2547 | 192 |
| 012 | realistic | 40 | 20 | 2547 | 384 |
| 013 | realistic | 20 | 20 | 2547 | 768 |

Figure 2: Time-mean vertical profiles of horizontal wind (a), its standard deviation (b), TI (c), resolved and total tke (d), and the ratio of resolved to subgrid-scale tke (e), for the different resolutions. The dots indicate model levels, and the rotor layer is shaded.

run, aerodynamic losses can be quantified.

Figure 4 shows probability density functions (PDFs) of wind at hub height in the center of the domain, and total power production, for all resolutions, and for the realistic and free stream run. The wind PDFs resemble a Weibull-shape, but due to the relatively short simulation period of 10 days, and the day-selection technique, it is not completely smooth. The power production PDFs peak at rated power. The PDFs of both wind and power production are similar for the different resolutions, which is reflected in the mean wind and power production values, shown in Table 3.2. The mean values in both total power production and wind differ by a few percent between the different resolutions. For the realistic runs, power production decreases with coarser resolution; whereas for the free-stream runs, it increases with coarser resolution. These opposing trends cause the aerodynamic losses to differ between the resolutions (ranging between 7 % and 11 %). A similar effect was reported in Baas et al. (2023).

However, besides this difference in mean value between the different resolutions, the general shapes of the wind- and power PDFs agree. This indicates that the basic physical processes that govern the lower-boundary layer and wind farm dynamics are well-represented at all presented resolutions.

To illustrate the effect of realistic inclusion of wind turbines in the simulation on the wind field, Fig. 5 shows maps of the mean wind, filtered to wind directions between 225°— 255°. Also, the velocity deficit with respect to the free-stream runs are shown. In general, there is a pronounced effect of the park on the

[Figure]

Figure 3: Mean diurnal cycles of (a) wind, (b) its standard deviation, (c) TI, and (d) resolved tke. The data are shown at the model levels closest to hub height, which is 75 m for the 120 m resolution run, and 70 m for the others.

Table 2: Mean total realistic and free stream power, their fraction, and mean realistic and free stream wind (at hub height, in the center of the domain), and their fraction.

| resolution (m) | mean realistic power (MW) | mean free-stream power (MW) | realistic power / free stream power (-) | mean realistic wind (m s$^{-1}$) | mean free-stream wind (m s$^{-1}$) | realistic wind / free stream wind (-) |
|---|---|---|---|---|---|---|
| 120 | 84.33 | 95.08 | 0.89 | 8.74 | 9.85 | 0.89 |
| 80 | 84.63 | 94.33 | 0.90 | 8.72 | 9.88 | 0.88 |
| 40 | 85.42 | 93.55 | 0.91 | 8.65 | 9.85 | 0.88 |
| 20 | 86.46 | 93.04 | 0.93 | 8.53 | 9.80 | 0.87 |

mean flow inside and downstream of the park. For the finest resolution, individual turbine wakes can be discerned, whereas for the coarser resolution, these individual wakes are less clear. Nevertheless, the general shape and magnitude of the wind park wake is captured by all resolutions.

**4 Conclusion**

In this supplementary study, the effect of LES resolution on lower boundary layer dynamics, and on the wind farm dynamics was shown. The resolution was varied between 20 m, enough to resolve individual turbines, to 120 m, the resolution used in the main text of this study.

Mean profiles of wind- and turbulence variables did not vary considerably between resolutions, and the differences between resolutions were smaller than the typical variations over the rotor layer. Also, differences between resolutions are likely smaller than the typical ASPIRE bias (for mean wind, this bias in the order of 0.1 m s$^{-1}$). The probability distribution of both wind and total power production were very similar between the resolutions, and the mean values of those quantities differed a few percent between resolution. The realistic and and free-stream runs show opposite trends with resolution, causing the aerodynamic losses to differ more between the different resolutions. This will remain an active area of research for the ASPIRE model.

[Figure]

Figure 4: PDFs of (a) wind at hub height in the center of the domain and (b) total power production. Vertical lines denote the mean values.

Finally, maps of mean wind speed show the wake effect. Although for individual turbines, the wakes are not resolved at the coarsest resolutions, the bulk effect of the wind farm on the flow field is very similar between resolutions.

To conclude, this study showed that the essential physics of boundary layer meteorology and wind farm dynamics can be sufficiently captured at resolutions that do not resolve individual turbine wakes.

[Figure]

Figure 5: Visualisation of wind park wakes at hub height for the different resolutions. Top row: maps of mean wind of the realistic runs, when the wind direction is in the sector 225°— 255°. Bottom row: velocity difference between the realistic and free-stream runs.

---

## Author Comment (AC4)

Dear editor,

We thank the reviewer for their comments, and we agree that the relevance and novelty of our study is the statistical long-term correction method, and not the use of LES for wind resource assessment in itself. Because the long-term correction method requires time series data, it can be applied to any model that produces time series. The benefits of such models are the possibility to compare to observations, and calculate correlations with e.g. electricity prices. LES is an example of such a model, that is gaining in popularity and usability in wind resource assessment for those reasons. This is why our paper presents the method using LES data. Nevertheless, we agree that it is but one example of a possible application. We have adapted the manuscript to emphasize this, and furthermore changed its title to: *Estimating Long-Term Annual Energy Production from Shorter Time Series Data: Methods and Verification with a 10-Year Large-Eddy Simulation of a Large Offshore Wind Farm.*

The reviewer's second point concerns the effects of other variables than wind on power production, which might be very pronounced in wind farms with irregular layouts. These effects can be quantified by the spread in wind farm power production values at a given wind speed, convoluted with the occurrence frequency of that wind speed. This spread, therefore, should be well-represented in the short term. In the example where a wind farm's power depends heavily on wind direction, this means that all wind directions should be represented. The error caused by the remaining misrepresentation can be quantified as a function of wind speed:

$$E(M) = \int ((\hat{h}_{\text{L} \mid \text{ERA}}(P, M) - h_{\text{L} \mid \text{ERA}}(P, M)) \hat{g}_{\text{ERA}}(M) P dP, \qquad (1)$$

in the notation of the manuscript, also see our reply to RC1. In the revised manuscript, have adapted its Fig. 4 to show $E(M)$ (see panel f in the figure below). Here, we see that the error is indeed mainly caused in the region where the wind farm power curve shows the largest spread. For irregularly spaced wind farms, which might have a larger spread in power given a certain wind speed, we can therefore expect a lower accuracy of the long-term correction method.

We think that showing the accuracy of the method for wind farms with different layouts is out of scope for our study. However, in the revised manuscript, we have elaborated more on this very valid point, and explained how irregularly spaced layouts might affect the accuracy of the long-term correction method.

[Figure]

The new version of Fig 4: Illustration of the long-term correction method. a) ERA5 100 $m$ wind distributions for 2010 and 2010-2019, and b) realistic LES power production distributions for 2010 and 2010-2019, and including the distribution for 2010-2019 as estimated by the long-term correction method from 2010. c) for all years 2010-2019 (increasing from blue to yellow) the long-term mean power as estimated by the long-term correction method, as a function of wind bin size. The chosen value of 0.75 $m$ $s^{-1}$ is indicated with the vertical line. d) for wind bins starting between 3 $m$ $s^{-1}$ and 12 $m$ $s^{-1}$ (indicated by the different colors, increasing from left to right), the power distribution within that wind bin for the years 2010-2019 (different lines). The long-term counterparts are plotted with dotted lines. f) the Perkins Skill Score between the short- and long-term power distribution within each wind bin and for each year. f) the error contribution (and its averaged absolute value) for each wind bin and for each year (same colors as in e).

---

## Author Response (AR1)

Dear editor,

Our manuscript received three useful reviews and one community comment. We uploaded Author Comments to each of them, containing detailed point-by-point responses and the subsequent changes made in the manuscript. In summary, the major changes made in the revised version are:

- The title, abstract, and introduction were rephrased to increase the emphasis on the long-term correction method in the broader context of wind resource assessment, rather than just in the LES context. We explained that LES is just one possible application of the presented methods, albeit a very suitable one.

- A paragraph on the position of our work in the broader literature about climate data downscaling was added.

- More background, references to earlier work, and explanation of the employed settings for the LES model were given.

- A resolution study was performed to show the effect of refining the LES resolution (it is attached to AC3), which could be added as supplementary material. However, we feel that it would not contribute very much to the narrative of the manuscript, which is presenting the long-term correction methods.

- To further quantify the validity of the assumption underlying the long-term correction method, an analysis of the Perkins Skill Score per wind bin, and the error contribution per wind bin was added, both in text and as additional panels to figure 4. An additional figure showing the error contribution per wind bin as a function of the number of days was added as well.

All things considered, we feel that the changes improved the clarity and scientific quality of the manuscript.

---

## Referee Report (RR1)

The revised manuscript describes and evaluates methods to long-term correct a time series of wind farm production from an offshore wind farm calculated with a high-fidelity model. The main approach is to use a probabilistic wind farm power curve derived from the shorter time series and to integrate over the joint probability functions of power curve and wind speed from the long-term data set to derive the long-term corrected mean energy production of the wind farm. In addition, the authors compare different simple day-selection methods for the high-fidelity time series and demonstrate how the results converge with the number of days used for the day-selection methods in isolation and in combination with the long-term correction method.

The revision of the manuscript makes the objective much clearer. Apart from some bad editing (e.g. duplicated words) which should be corrected, my main points of criticism on the revised manuscript are:

- The authors mention that the method is applicable if the probabilistic wind farm power curve derived from the short-term time series approximates the "true" long-term power curve (equation 7). The distribution in each wind speed bin represents the influence of other meteorological measures like most importantly the wind direction. For the probabilistic power curve of the short-term sample to be close to the long-term equivalent, the distributions of these other meteorological measures should be similar to their long-term distributions. For a symmetric wind farm the wind speed distribution might be sufficient to describe the long-term mean, as evident by the good performance of the *ordered* method to select the sampled days even without using a the long-term correction. But most wind farms are not symmetric and have wind directions in which the wake losses are clearly higher than for other. The described methods provide no mean to evaluate this; thus the method should only be considered as suitable for wind farms with a layout for which the wake losses are mostly independent of wind direction.

- (l. 200) A 0.35 m/s difference is not a small bias in the context of wind farm siting at a site with a mean wind speed of around 9 m/s. This corresponds to a bias of gross power of at least 5%. Also, the relative sentence "which is typical in this application" needs more context. If a wind speed bias of 4% is typical for LES, then it is worse than typical biases reported from e.g. mesoscale simulations (Hahmann et al. https://doi.org/10.5194/gmd-13-5053-2020).

I think the manuscript is worth publishing if the authors clearly mention the shortcomings listed above in their conclusion.

Other comments:

- L. 235 - "involved techniques" - you probably mean "evolved"
- L. 337 ff. - Labelling the MCP-corrected wind speed "MCP wind" seems odd. Please find a different name.

---

## Author Response (AR3)

Dear editor,

We are pleased that the final review is there. A point-by-point reply:

- We agree with the reviewer that the manuscript does not evaluate the wind direction (or other) dependence of power production. We therefore added the following sentence in the conclusion: "The performance of the methods depends on the validity of the main assumption behind them: that the short-term power distribution within each wind bin can approximate its long-term counterpart. Or equivalently, that all meteorological factors that control power production of a large wind farm (wind, wind direction, stability, air density, ...) are similarly distributed in the short-term and long-term. The accuracy of this assumption might be different in other wind farms, for examples ones that show a strong directionality because of their shape. The errors reported in this study are therefore indicative, and cannot be readily translated to all other situations.".

- The reviewer raises a valid point. We choose to refrain from any value judgement about the magnitude of the bias, but state it as is. Therefore, we removed the relative clause 'which is typical in this application'.

- We do mean involved, as in complex, complicated, sophisticated. See `https://www.merriam-webster.com/thesaurus/involved`.

- We changed the name to 'MCP-corrected wind'.